# SENSE IT WITH YOUR EYES: SENSATION GENERATION AND UNDERSTANDING FOR ADVERTISEMENTS

## ABSTRACT

Effective advertisements might persuade the audience by evoking human sensations, yet current Text-to-Image (T2I) models struggle to generate persuasive visuals that convey implicit sensory experiences. We introduce the **SensoryAd Generation task**: given an advertisement message and a specific sensation related to the advertisement message, the goal is to generate advertisement images that both convey the message and evoke the sensation. To support this task, we build the SensoryAd dataset, consisting of human-designed and generated advertisements annotated with sensation categories, visual elements evoking the sensation, and human ratings. We further propose an evaluation method using contrastive and consistency losses across hierarchical sensation levels.

## 1 INTRODUCTION

> "I have left behind illusion, I said to myself. Henceforth I live in a world of three dimensions–with the **aid of my five senses**. I have since learned that there is no such world, but then, as the car turned out of sight of the house, I thought it took no finding, but lay all about me at the end of the avenue."
>
> *Evelyn Waugh, "Brideshead Revisited"*

The full spectrum of senses (not only vision and hearing, but also touch, smell and taste) is important for humans to navigate and experience their environments. However, humans sometimes hallucinate sensations, with very real effects: people experiencing lexical-gustatory synesthesia experience taste triggered by words (Ward & Simner, 2003), visually impaired people can "see" with their tongue through electrical signals (Nau et al., 2015), phantom limb pain can be treated with augmented reality (Prahm et al., 2025), and advertisements (ads) can evoke taste (Palcu et al., 2019).

An effective ad is not only defined by what elements it represents, but also by *how* they are represented. Designers often rely on creative techniques to better capture attention and enhance credibility and impact of ads. Sensory advertising (Krishna, 2012) is one creative technique, where some ads evoke one or more of the five human senses (e.g., touch, taste), allowing the audience to imagine the benefit of a product or the consequence of an action in a visceral way. Stimulating the senses in the exact sense modality is infeasible (e.g., through the taste buds) so ads resort to visual content *associated* with the target sensation. For example in Fig. 1, for a beer advertisement, on a hot summer day, image (b) is more likely to convince a thirsty audience to buy the beer by evoking the cooling and refreshing sensation (through the inclusion of the ice cubes), compared to image (a). Similarly, evoking the pain sensation in image (d) makes the parents better feel the consequence of using negative words by feeling the pain, compared to the more sensory-neutral image (c).

In this work, we conduct the first investigation of *how ads evoke the senses through visual means*. We focus on three facets, (1) understanding, (2) evaluating and (3) generating sensory ads. First, we develop a taxonomy of senses at different levels of granularity in which the first layer corresponds to these five fundamental sensory modalities (information perceived through the five primary human sensory organs of eyes, ears, nose, skin, and tongue). These senses are then further refined into more specific subcategories (e.g., "temperature" is a type of "touch" sensation). Fig. 1, e and g, represent example outputs of T2I models on given "cooling and refreshing" and "sharp pain" sensations. We construct a dataset with samples of these senses, by collecting annotations from Prolific annotators on 670 images sourced from an existing dataset of advertisements (PittAd (Hussain et al., 2017)). We

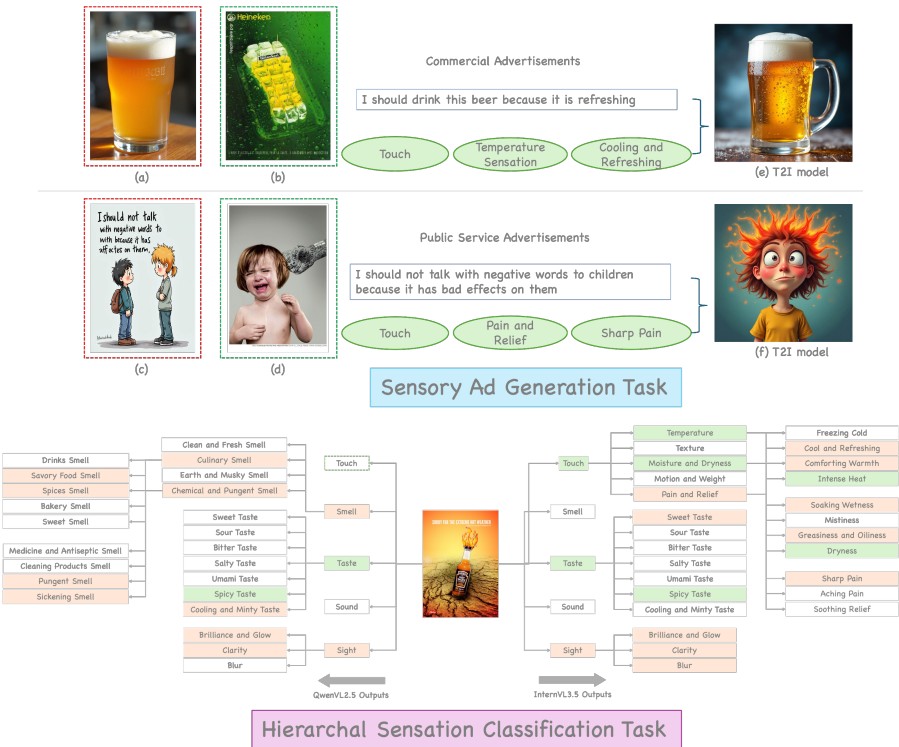

Figure 1: **Sensory Ad Generation Task (Top):** Two examples of ad images that evoke desired sensation for action-reason statements (b) and (d) designed by human, and images that do not evoke any sensation (a) and (c) generated by T2I models without sensory awareness. Images (e) and (f) represent the output of T2I models on the SensoryAd generation task given the ground-truth sensations for the advertisements. **Hierarchical Sensation Classification Task (Bottom):** Image shows an example of two MLLMs on the proposed task given an image evoking the sensations of spicy taste, intense heat, dryness. In the image, green background represents correct sensations chosen by the model, and red background represents sensations chosen incorrectly.

collect information about whether the image evokes a sensation and if so, the category of sensation, the visual elements evoking it, and score of how well the image evokes the sensation.

We introduce two sensation classification tasks to evaluate how well LLMs and MLLMs perform on the task of classifying the senses in an ad.

Second, we propose a novel evaluation method, **EvoSense**, that measures how effectively an image evokes a target sensation. EvoSense first utilizes an LLM to generate the description of the image and then use a fine-tuned LLM to get the average log probability of the tokens of the target sensation when prompting the model with *"The described image evokes the: "*. Experimental results show that our evaluation metric achieves a Kappa (Cohen, 1960) agreement score of 0.86 with human annotators, representing an improvement of 56% over existing baseline metrics.

Third, we introduce the **SensoryAd Generation** task, where the goal is to generate advertisement images that both convey a given message and evoke a specified sensation. The messages are collected from the PittAd dataset (Hussain et al., 2017) and structured in the form *"I should {action} because {reason}"* called action-reason (AR) statements. Our results show that existing T2I models fail in generating advertisement images that evoke specific sensation.

To summarize our contributions: (1) We introduce the SensoryAd dataset including the sensations advertisement images evoke, the score on how well the images evoke each sensation, and visual elements in the image evoking the sensation. (2) We introduce two sensation classification tasks. (3)

We introduce the novel task of Sensory Ad Generation. (4) We propose an evaluation method for sensation evocation in generated images.

## 2 RELATED WORKS

**Text to Image Generation.** T2I models such as Flux (Black Forest Labs, 2024), Stable Diffusion (Esser et al., 2024), Qwen-Image (Wu et al., 2025), PixArt (Chen et al., 2024), DALLE3 (Betker et al., 2023), etc. have advanced in generating high quality and realistic images given the explicit description of the prompt. Some existing work (Aghazadeh & Kovashka, 2024; Liao et al., 2024) assess the capability of models in generating images from abstract concepts and messages like advertisement design tasks. (Yang et al., 2024b; Dang et al., 2025; Park & Lee, 2020), focus on emotion transfer through images. The main focus of these works is on transferring emotion which is the interpretation of sensation and differs from evoking sensation. For example, in Fig. 1 both image (c) and (d) can transfer sadness, but only image (d) evokes the pain sensation. In this work, we benchmark the T2I models on generating advertisement images that evoke specific sensations.

**Text to Image Evaluation.** Existing evaluation metrics, such as (Lin et al., 2024; Xu et al., 2023), are primarily designed to assess how well an image corresponds to an explicit prompt. These metrics are effective when the prompt specifies concrete objects, attributes, or relations between visual elements. Evaluating **sensation evocation** poses a unique challenge: the sensation is not only an implicit concept but the same sensation can be represented through entirely different visual designs.

**Understanding Modalities beyond Sight.** Our work is part of a bigger trend of understanding modalities beyond sight, including understanding audio and touch data (Ghosh et al., 2024; Yang et al., 2024a) or semantic-taste mappings using the wine taste dataset of (Bender et al., 2023). Other work seeks to predicting physical properties such as density and hardness from images and descriptions (Zhai et al., 2024). However, no prior work studies how images are created to evoke specific sensations, nor predicts computationally the impact of sensations on an audience.

**Understanding and Generating Advertisements.** Hussain et al. (2017) pioneer the task of computational visual ad understanding, but do not capture sensory information. Prior work has investigated the use of T2I models for generating advertisements, focusing on criteria such as creativity, and persuasion (Aghazadeh & Kovashka, 2024) or for depicting specific metaphorical relationships (Akula et al., 2023). However, these studies do not examine the models' ability to implement specific persuasion strategies, such as the evocation of specific sensations, which play a crucial role in making advertisements more influential and memorable.

**Sensory Advertising.** (Krishna, 2012) define sensory marketing as "marketing that engages the consumers' senses and affects their perception, judgment and behavior." The author describes evidence that the subconscious sensory triggers may make the ad's message more compelling than explicit messaging, including causing viewers to perceive specific properties of the product. They discuss the sensory aspects of product packaging (e.g., Hershey's chocolate kisses creating the sensation of a drop melting), sound symbolism (e.g., the word "frosh" evoking the sensation of creaminess more than "frish"), the memories scents create and evoke, etc. Cian et al. (Cian et al., 2014) describe the dynamics encoded in similar but slightly varied imagery (e.g., a horizontal vs tilted seesaw). Other related work in marketing and psychology studying sensory marketing is (Krishna & Schwarz, 2014; Petit et al., 2019; Hultén, 2015; Krishna et al., 2016). Related to sensory image Yang et al. (2024b) focus on emotion, interpretation of sensation by human, generation and Singh et al. (2024) on understanding content with focus on human reaction upon receiving content.

## 3 SENSORYAD BENCHMARK

### 3.1 SENSORYAD DATASET

**Sensation Hierarchy (Taxonomy).** Some advertisements are designed to evoke sensations that help the audience imagine a specific situation and the need for a product more vividly, which is an important factor in ad effectiveness (Krishna et al., 2016). In this work, we formalize the notion of sensation using a hierarchical taxonomy (partly shown in Fig. 2; complete hierarchy in sec. A.4). At the top level, our taxonomy corresponds to the five primary human senses. Each of these is

Figure 2: **Sensation Hierarchy** (left; only categorizing the Touch sensation): Green box represents first level sensation, blue represents second level, and purple represents third level sensations. **Annotation Example** (right): Example of annotations in our SensoryAd dataset.

further subdivided into more fine-grained categories. For example, "Touch" is refined into "Texture", "Temperature," "Moisture and Dryness," "Pain and Relief," and "Pressure". By definition, if an image evokes a child sensation (e.g., "Temperature"), it also evokes its parent (e.g., "Touch"). We introduce a dataset of both real and generated ads annotated with (i) up to three groups (leaf and ancestors) of sensations evoked by each image, (ii) a score reflecting the strength of evocation, and (iii) the visual elements that contribute to the sensation. Fig. 2 shows an example annotation.

**Data Collection.** We first selected 670 images from the PittAd dataset (Hussain et al., 2017), including 250 public service advertisements (designed to raise awareness about societal issues or influence behavior) and 420 commercial advertisements (promoting products or services) to ensure a diverse range of sensory content. We have included the data statistics including the topics diversity, sensations diversity, and human-human agreement in sec. A.4. Annotation was carried out by trained crowd-workers on Prolific and using forms created on Qualtrics. Before contributing, each annotator was tested and approved/filtered based on completing a practice form after reading detailed instructions, definitions of sensations, and illustrative examples. The annotation task followed a structured protocol: annotators first chose the most prominent sensation among the five top-level categories (with the option of selecting "None" if no sensation was evoked). Based on their choice, they were presented with progressively narrower subcategories until reaching a leaf-level sensation. For each selected sensation, annotators provided a strength score and listed the visual elements (e.g., colors, objects, textures) that contributed to it, using free-form text (which can be used in future work). This process was repeated up to three times per image unless "None" was chosen as the sensation evoked by the image. We also get the human-human agreement on about 10% of the annotated images and the Kappa agreement Cohen (1960) is 0.83 with 95% confidence interval of [0.831, 0.838]. The full annotation and testing forms are provided in the supplement file, and the dataset will be released upon acceptance.

We also annotated ad images generated by text-to-image models, to test performance of our evaluation model on these. First, we used the action-reason statements (from (Hussain et al., 2017)) and three annotated sensations (from the above paragraph), as inputs to three T2I models: Flux (Black Forest Labs, 2024), AuraFlow (Fal, 2024), PixArt (Chen et al., 2024), Stable Diffusion 3 (Esser et al., 2024), and Qwen-Image (Wu et al., 2025). From 75 images generated by each model, we randomly selected 15 and annotated them using the same procedure as for the real ads.

## 3.2 SENSATION CLASSIFICATION TASK

Interpreting sensory ads, and the evaluation of their effectiveness, hinges on understanding which sensations an image evokes and with what intensity. Moreover, Some sensations like pain sensation can be sensitive to a group of audience such as children in a certain age. Given this, to prevent the presentation of a specific sensation to a specific group, the filtering systems should be able to detect the sensations evoked by the content. To formalize this, we introduce the **Sensation Classification Task**, which involves recovering the correct sensations evoked by an image. We consider two complementary formulations: (i) **Hierarchical Selection** and (ii) **Single Selection**.

**Hierarchical Classification.** In this setting, sensations are defined according to our hierarchical taxonomy. Data annotation proceeds level by level: starting from the top-level categories, annotators choose the sensation best evoked by the image, then move to its subcategories, and so on until reaching a fine-grained leaf. The Hierarchical Selection Sensation classification task mirrors this process. Given an image, the goal is to predict the complete sensation path(s) from the root to the

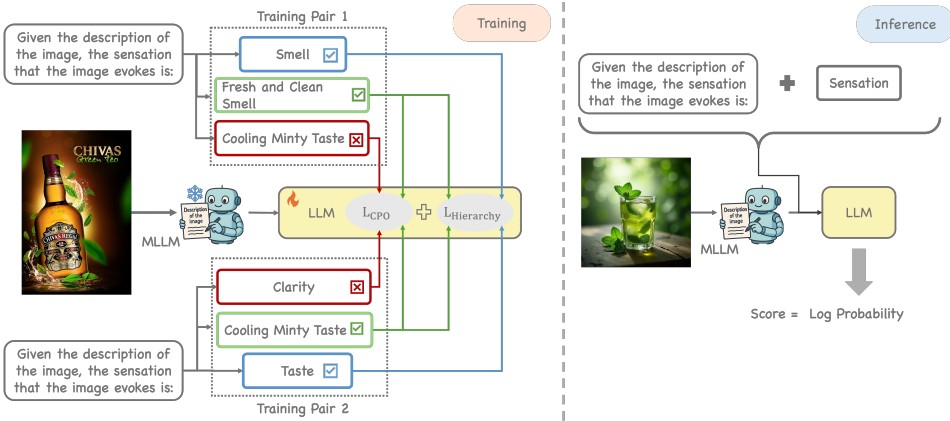

Figure 3: **EvoSense Evaluation Method.** Left: training of LLM with two different set of sensations for one image. Green border shows the winner sensation, blue border represents the parent of winner sensation (used in hierarchy loss), and red border denotes the loser sensation in the pair. Each pair is derived from a triplet of annotations, where A is preferred over B, and B over C. Right: score computation in inference with the fine-tuned LLM.

leaf node (e.g., Touch → Temperature → Freezing Cold). To do so, a model is recursively prompted to predict up to three sensations, advancing down the hierarchy by selecting among the children of each of the previously chosen nodes. To provide sufficient context, the definition of each potential sensation was included in the prompt. An example of this task is visualized in Fig. 1.

**Single (Multi-Label) Classification.** This task flattens the taxonomy and treats every sensation, regardless of its level, as a potential label. The goal is for a model to predict the complete set of sensations that an image evokes from all possible labels in the hierarchy. A critical constraint in this task is maintaining hierarchical consistency. By definition, if an image evokes a specific sensation, it must also evoke its parent sensation (e.g., if "Temperature" is evoked, "Touch" is evoked as well). To evaluate a models' understanding of these relationships, we define an additional metric: Parent Recall ($R_{parent}$), which measures the fraction of predicted non-root sensations for which the direct parent sensation was also predicted. It is formally defined as:

$$R_{parent} = \frac{|\{s \in S_{pred} \mid s \text{ is not a root node and } parent(s) \in S_{pred}\}|}{|\{s \in S_{pred} \mid s \text{ is not a root node}\}|} \quad (1)$$

where $S_{pred}$ is the set of sensations predicted by the model. A high $R_{parent}$ score indicates that the model understands the hierarchical dependencies of sensations.

## 3.3 EVOSENSE: EVALUATING SENSATION EVOCATION

Sensation evocation can make advertisements more persuasive by enabling viewers to vividly picture the intended context. To quantitatively assess this effect, it is not sufficient to simply identify which sensations are present; it is also crucial to evaluate their intensity. To address this, we introduce **EvoSense**, an evaluation method to assess how strongly an image evokes a given sensation. EvoSense uses two stages. (i) **Image Description Generation:** An MLLM (e.g., InternVL) generates a textual description of the image. (ii) **Sensation Intensity Scoring:** An LLM is prompted with the template *"Given the description of the image, the sensation that the image evokes is: "* and the average log-probability of producing the target sensation is reported as the sensation intensity score.

Initial experiments using zero-shot LLMs show low agreement with human annotations, both in retrieving correct sensations and in estimating their intensity. To address this, we fine-tune the models using a subset of our annotated dataset. In our task some sensations are evoked more than some other sensations, for example, in Fig. 2 while both *Taste* and *Smell* are evoked by the image, *Smell* sensation is evoked more. A simple supervised fine-tuning approach treats both sensations as equally correct. In contrast, by pairing sensations and asking the model to choose, *Smell* should be preferred over *Taste*. On the other hand, when *Taste* is paired with *Sight*, *Taste* should be preferred

instead. To capture such relative preferences, we train EvoSense with **Contrastive Preference Optimization** (CPO) (Xu et al., 2024) loss, which requires models to rank sensations. However, CPO does not account for the hierarchical dependencies between sensations (i.e., a child sensation implies its parent). To incorporate this structure, we augment CPO with a hierarchy loss:

$$
\begin{aligned}
L_{\text{CPO+Hierarchy}} = &-\log \sigma \big( \beta \big[ \log \pi_\theta(y^+ \mid x) - \log \pi_\theta(y^- \mid x) \big] \big) \\
&+ \text{ReLU} \big( \log \pi_\theta(y^+ \mid x) - \log \pi_\theta(y^{\text{parent}} \mid x) \big) .
\end{aligned}
\tag{2}
$$

where $x$ is input (prompt), $y^+$ is preferred output, $y^-$ is rejected output, $y^{parent}$ is parent of chosen output, $\pi_\theta(y \mid x)$ is the model's conditional probability of $y$ given $x$, $\beta$ is a temperature scaling factor, and $\sigma(\cdot)$ is the logistic sigmoid function. $L_{CPO+Hierarchy}$ encourages the model to choose $y^+$ over $y^-$ and prevent the probability of $y_{parent}$ to be lower than $y^+$. We illustrate in Fig. 3.

### 3.4 SENSORYAD GENERATION TASK

Recent advancements in Text-to-Image (T2I) generation have enabled the generation of high quality and realistic images, leading to their adoption in applications such as automated advertisement image generation. While prior work has studied the ability of generative models to convey emotions Yang et al. (2024b), their capability in generating images that evoke specific sensations, which is a persuasive strategy, remains unexplored.

To address this gap, we introduce the **SensoryAd Generation** task where the input consists of an advertisement message (action-reason statement (Hussain et al., 2017)) and target sensation, and the objective is to generate an image that effectively evokes the specified sensation. Examples of outputs from existing T2I models for different prompts and target sensations are shown in Fig. 1 (e, f).

## 4 RESULTS

This section presents our experimental results. We begin by benchmarking LLMs and MLLMs on our sensation classification tasks to assess their understanding of sensory concepts. We then validate our proposed EvoSense metric, comparing against baseline metrics. Finally, we evaluate the performance of leading T2I models on the SensoryAd Generation task. Implementation details for all experiments are in the sec. A.5.

### 4.1 SENSATION CLASSIFICATION TASKS

We assess the capability of LLMs and MLLMs on our two sensation classification tasks. Our evaluation follows two distinct protocols based on the model's input modality. For MLLMs (InternVL, QwenVL, and Gemma, and LLAVA-Next), the advertisement image was provided as direct visual input. The model was then tasked with classifying the corresponding sensations based on a task-specific prompt (see sec. A.5). To assess the performance of text-only LLMs (LLAMA3-instruct, QwenLM, and Gemma), we employed a two-stage pipeline. First, we utilized different MLLM (InternVL, QwenVL, and Gemma) to generate a description for the image ($D_{MLLM}$). These generated descriptions were utilized as input context for the LLMs to perform sensation classification. This approach allows us to isolate and evaluate the language-based reasoning capabilities of LLMs for this specific task. We report Recall (R), Precision (P), and F1-score (F1). For Single Classification, we also report the Parent Recall ($R_{parent}$) to assess understanding the hierarchical relations.

**Hierarchal Classification.** Table 1 reveals a consistent trend across all models: significantly higher recall than precision. This imbalance indicates that while models are proficient at identifying potentially relevant sensations, they struggle to reject incorrect ones, leading to many false positives. Notably, MLLMs generally outperform their LLM counterparts on this task. This suggests that direct visual input provides crucial cues that are lost or distorted in text-only descriptions. This loss is particularly evident in the performance of Gemma (Team et al., 2025). The MLLM version, which processes the image directly, achieves higher Recall and F-1 score than the LLM version, which operates on a text description from that same MLLM.

**Single Classification.** The results in Table 1 show that while MLLMs achieve higher precision and F1-scores, LLMs have a stronger performance on Parent Recall ($R_{parent}$). This suggests MLLMs are more adept at grounding their selections in visual evidence, leading to more accurate overall

| Model | Hierarchal Selection | | | Single Selection | | | |
|---|---|---|---|---|---|---|---|
| | $P$ | $R$ | $F1$ | $P$ | $R$ | $F1$ | $R_{parent}$ |
| **MLLMs** | | | | | | | |
| QwenVL | 0.17 | 0.62 | 0.10 | **0.33** | 0.18 | **0.11** | 0.45 |
| InternVL | 0.13 | 0.60 | 0.08 | 0.18 | 0.44 | 0.08 | 0.41 |
| LLAVA-Next | 0.10 | 0.60 | 0.07 | - | - | - | - |
| GEMMA | 0.17 | **0.66** | **0.11** | 0.11 | 0.39 | 0.05 | 0.49 |
| **LLMs** | | | | | | | |
| QwnLM + $D_{QwnVL}$ | 0.18 | 0.45 | 0.08 | 0.18 | 0.42 | 0.07 | 0.24 |
| GEMMA + $D_{QwenVL}$ | 0.16 | 0.54 | 0.09 | 0.13 | **0.54** | 0.07 | **0.65** |
| LLAMA3 + $D_{QwenVL}$ | 0.19 | 0.43 | 0.08 | 0.15 | 0.47 | 0.08 | 0.48 |

Table 1: **Sensation Classification:** Results of MLLMs and LLMs on classification tasks.

| Metrics | Real Ads | | Generated Ads | |
|---|---|---|---|---|
| | $r$ | $\kappa$ | $r$ | $\kappa$ |
| VQA-score | 0.27 | 0.55 | 0.25 | 0.52 |
| Image-Reward | 0.21 | 0.46 | 0.21 | 0.40 |
| CLIP-score | 0.22 | 0.43 | 0.21 | 0.45 |
| Pick-score | 0.15 | 0.38 | 0.15 | 0.41 |
| LLAMA3-instruct (zero-shot) + $D_{InternVL}$ | -0.02 | -0.01 | -0.02 | -0.01 |
| QwenLM (zero-shot) + $D_{InternVL}$ | -0.02 | -0.02 | -0.02 | -0.04 |
| EvoSense (LLAMA3-instruct + $D_{InternVL}$) | **0.38** | **0.86** | **0.31** | **0.68** |
| EvoSense (LLAMA3-instruct + $D_{QwenVL}$) | 0.35 | 0.80 | **0.31** | 0.67 |
| EvoSense (QwenLM + $D_{InternVL}$) | 0.32 | 0.70 | 0.26 | 0.56 |
| EvoSense (QwenLM + $D_{QwenVL}$) | 0.30 | 0.65 | 0.26 | 0.55 |

Table 2: **Metric Quality.** Pearson Corr. ($r$) and Kappa agreement ($\kappa$) between metric [scores/chosen sensations] and human [scores/chosen] on 5000 real and 5000 generated image-sensation pairs.

classification. Conversely, LLMs, operating on textual descriptions and definitions, develop a better understanding of the abstract, semantic relationships between sensations in the hierarchy.

## 4.2 EVOSENSE

**EvoSense** is our evaluation method designed to assess the intensity with which an image evokes a specific sensation. To evaluate the accuracy of our metric, we use human annotations in our dataset (separate from those used for training EvoSense). For each image, if a sensation was chosen by an annotator, the intensity of that sensation is set the same as the score chosen by the annotator, otherwise it is set to zero. For agreement computation we use: (1) Pearson Correlation ($r$) between the scores for each (image, sensation) pair chosen by annotators and computed by the metrics, and (2) Kappa ($\kappa$), where we use the sensation with higher score as the chosen one.

**EvoSense compared to baselines.** We benchmark EvoSense against baseline metrics, including VQA-score (Lin et al., 2024), ImageReward (Xu et al., 2023), CLIP-score (Hessel et al., 2021), and Pick-score (Kirstain et al., 2023). To demonstrate the necessity of our proposed fine-tuning procedure, we further compare EvoSense against the zero-shot performance of the EvoSense inference pipeline using LLAMA3-instruct ($LLAMA3$) and QwenLM ($QwenLM$) with image descriptions generated by InternVL ($D_{InternVL}$) and QwenVL ($D_{QwenVL}$). As observed in Table 2, among baseline metrics, VQA-score achieves the highest human-metric agreement with moderate performance ($\kappa = 0.55$, $r = 0.27$ on real ads and $\kappa = 0.52$, $r = 0.25$ on generated images). In contrast, fine-tuned EvoSense reaches near-perfect agreement with human ($\kappa = 0.86$, $r = 0.38$) on

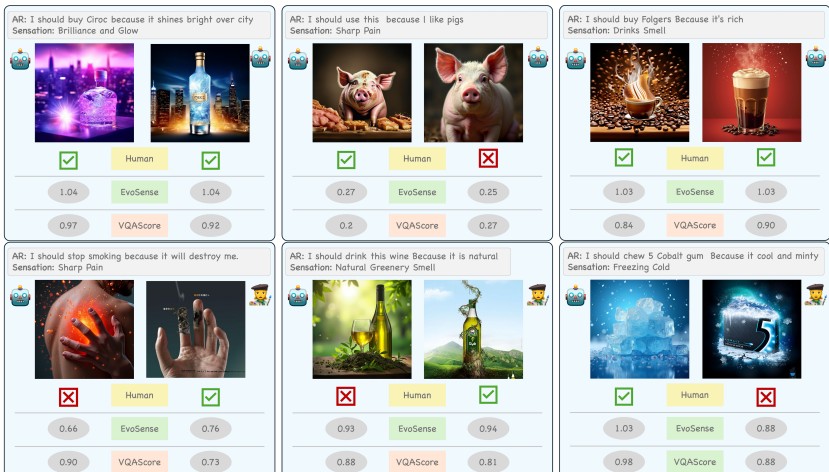

Figure 4: **Examples on human agreement with EvoSense and VQA-score** on intensity of sensations in the image. The Human row shows the chosen (✓) image(s) (including ties) and rejected (×) image. Red background indicates the model-chosen (higher-scoring) option is misaligned with human choice, and green background shows it is aligned.

real ads and substantial performance ($\kappa = 0.68$, $r = 0.31$) on generated ads, representing a 56% and 30% improvement on Kappa agreement for real and generated ads, and 41% and 24% on Pearson Corr. improvement over the best baseline. Notably, zero-shot EvoSense exhibits negative agreement—complete misalignment with human judgments, emphasizing that our fine-tuning procedure is essential for alignment with human perception. The examples in Fig. 4 show higher agreement of EvoSense with human annotation compared to VQA-score (the best baseline), especially in cases where humans give equal scores for both images.

**Ablation on EvoSense.** We conducted an ablation study to analyze the impact of the core components of EvoSense: the base LLM and the MLLM used for description generation. The results in Table 2 show that while both fine-tuned LLMs significantly outperform all baseline metrics, LLAMA3-instruct holds a slight edge over QwenLM in human agreement. Furthermore, the results demonstrate the robustness of our method to the source of the image descriptions. When the descriptions are generated by QwenVL ($D_{QwenVL}$) instead of InternVL ($D_{InternVL}$), the change in agreement scores for the fine-tuned models is minimal.

### 4.3 SENSORYAD GENERATION

First, we benchmark different T2I models including Flux (Black Forest Labs, 2024), Stable Diffusion 3 (SD3) (Esser et al., 2024), AuraFlow (Fal, 2024), PixArt (Chen et al., 2024), and Qwen-Image (Wu et al., 2025), on the SensoryAd task evaluating their abilities in generating images that convey specific ad messages and evoke the given sensation to make the images more persuasive. Next, we benchmark the models on generating images that evoke specific sensation without any other information (such as ad message) in the prompt ('Generate an image evoking {sensation}'), to better understand their abilities in sensory image generation, as a reference point for our analysis.

**Sensation Intensity in Generated Ads.** Table 3 shows that among T2I models, Qwen-Image achieves the highest sensation intensity and SD3 has the lowest intensity. We note that while the goal is to evoke specific sensations, sometimes models exaggerate in evoking the sensation, overlook the advertisement message, and only show sensation-associated objects. For example, in Fig. 4, the image generated for an ad conveying *'I should chew 5 Cobalt gum Because it cool and minty'* evoking *Freezing Cold* sensation, only depicts ice-cubes in the image which does not convey the message. See sec. A.1 for further analysis comparing sensation intensity in generated and real ads.

**Comparison of Sensory Ad Generation and Sensory Image Generation.** Table 3 shows that sensation intensity in images (*not* ads) generated for *'Generate an image evoking {sensation}'* is

| T2I model | Sensory Ad Generation | | Sensory Image Generation | | | |
|---|---|---|---|---|---|---|
| | EvoSense (LLAMA3) | AIM | EvoSense (LLAMA3) | | EvoSense (QwenLM) | |
| | $D_{InternVL}$ | $D_{InternVL}$ | $D_{QwenVL}$ | $D_{InternVL}$ | $D_{QwenVL}$ | |
| Flux | 0.90 | 0.60 | 0.72 | 0.72 | 0.71 | 0.71 |
| SD3 | 0.89 | 0.61 | 0.69 | 0.68 | 0.68 | 0.69 |
| AuraFlow | 0.90 | 0.59 | 0.74 | 0.74 | 0.74 | 0.73 |
| PixArt | 0.91 | 0.59 | 0.76 | 0.75 | **0.76** | **0.76** |
| Qwen-Image | **0.93** | 0.63 | **0.77** | **0.76** | 0.75 | 0.75 |

Table 3: Evaluating Generated Sensory Ads, and Sensory Images

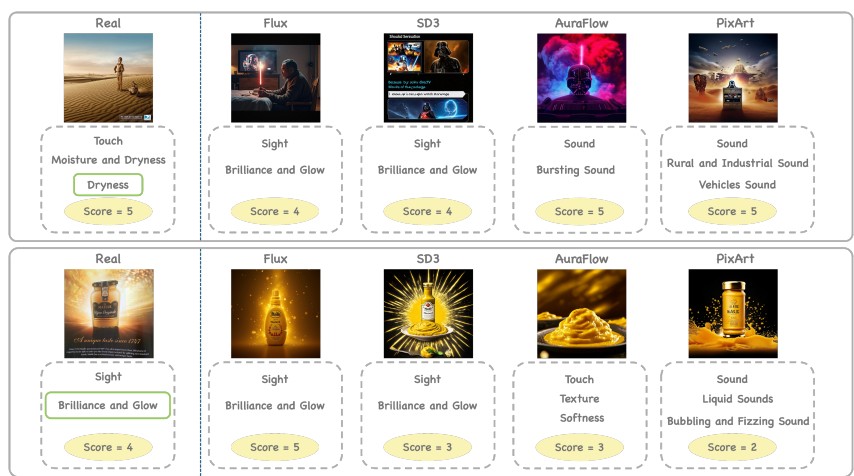

Figure 5: **Sensory Ad examples.** Two examples of real advertisement and generated advertisements by Flux (Black Forest Labs, 2024), SD3 (Esser et al., 2024), AuraFlow(Fal, 2024), and PixArt (Chen et al., 2024) given the action-reason interpretation and sensation annotation for the real advertisement. Green border represents the sensation used in the prompt of T2I models.

lower than intensity of sensation in Sensory Ads. When sensations are commonly associated with specific objects, the model exaggerates in evoking the sensation and overlooks the advertisement message, but when sensation is less common, or it is not associated with an object, existence of some visual elements or attributes in the advertisement message (action-reason statement) can help the model in evoking the sensation.

We compare intensity of sensations for real and generated images, and for different sensations, in Fig. 5 and sec. A.1. For some sensations like "Brilliance and Glow" which are either visual sensation or commonly associated with specific objects, not only can the model evoke the target sensation, but it can evoke it with higher intensity than the corresponding real image (ex. Flux in evoking Brilliance and Glow). In contrast, for sensations which are less visual, like Dryness, the models fail in generating images evoking the sensations.

## 5 CONCLUSION

We addressed the challenging, previously unexplored task of generating and understanding visual content that evokes specific human sensations, a crucial element of persuasive advertising. To facilitate research in this area, we introduced the *SensoryAd benchmark* including the SensoryAd dataset with a detailed hierarchical taxonomy for sensation, two Sensation Classification tasks, and the new SensoryAd Generation task. We propose *EvoSense*, an evaluation metric that accurately measures the intensity of evoked sensations. By fine-tuning an LLM with a hybrid objective (CPO and hierarchical constraints), EvoSense achieves high agreement with human judgments, significantly outperforming existing baselines by up to 56%. This work lays the foundation for developing a new generation of sensation-aware models and expanding the scope beyond advertising.

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

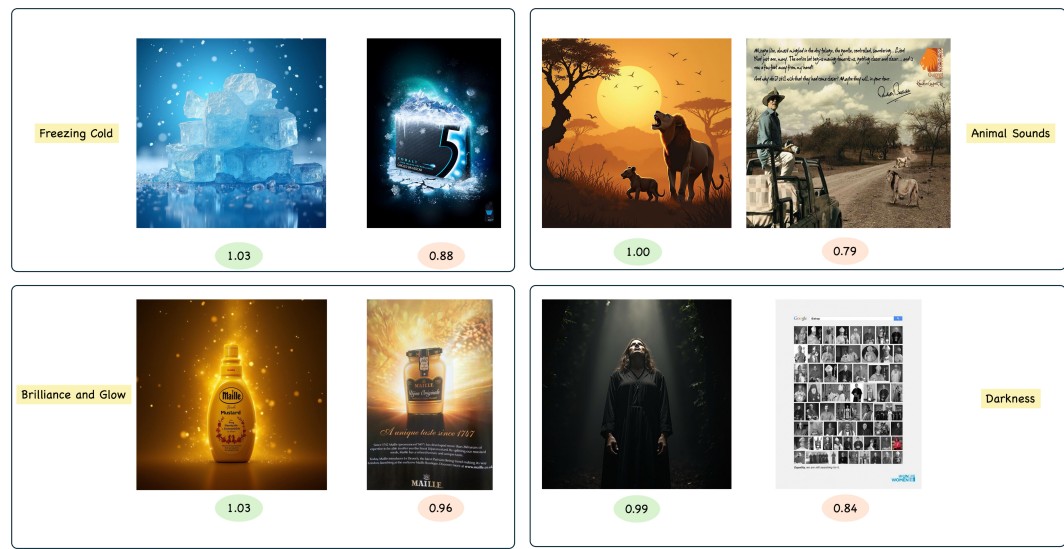

Figure 6: **Examples of exaggeration in sensation evocation.** In each group image on the left represents the generated images and image on the right represents real advertisement.

Jiazheng Xu, Xiao Liu, Yuchen Wu, Yuxuan Tong, Qinkai Li, Ming Ding, Jie Tang, and Yuxiao Dong. Imagereward: Learning and evaluating human preferences for text-to-image generation. *Advances in Neural Information Processing Systems*, 36:15903–15935, 2023.

Fengyu Yang, Chao Feng, Ziyang Chen, Hyoungseob Park, Daniel Wang, Yiming Dou, Ziyao Zeng, Xien Chen, Rit Gangopadhyay, Andrew Owens, et al. Binding touch to everything: Learning unified multimodal tactile representations. In *Proceedings of the IEEE/CVF Conference on Computer Vision and Pattern Recognition*, pp. 26340–26353, 2024a.

Jingyuan Yang, Jiawei Feng, and Hui Huang. Emogen: Emotional image content generation with text-to-image diffusion models. In *Proceedings of the IEEE/CVF Conference on Computer Vision and Pattern Recognition*, pp. 6358–6368, 2024b.

Sung-Joon Yoon and Ji Eun Park. Do sensory ad appeals influence brand attitude? *Journal of Business Research*, 65(11):1534–1542, 2012.

Albert J Zhai, Yuan Shen, Emily Y Chen, Gloria X Wang, Xinlei Wang, Sheng Wang, Kaiyu Guan, and Shenlong Wang. Physical property understanding from language-embedded feature fields. In *Proceedings of the IEEE/CVF Conference on Computer Vision and Pattern Recognition*, pp. 28296–28305, 2024.

# A APPENDIX

## A.1 COMPARISON OF REAL AND GENERATED ADVERTISEMENTS

As part of our analysis on how well T2I models are capable in generating effective intensity of sensation, we compared the intensity of sensation in generated advertisements and real advertisements. Average sensation intensity of sensations in real ads computed by EvoSense is 0.83 which is lower than intensity of sensations in the ads generated by T2I models. However, this does not represent the capability of T2I models in generating Sensory Ads, in contrast it is due to exaggeration in evoking sensations (Sensory Exaggeration) that are either related to visual sensations (any sensation under sight category) or sensations that commonly associated with some objects like 'Freezing Cold' which is commonly represented by ice-cubes or intense snow. In Fig. 6, generated images evoke the input sensation with higher intensity than real advertisement; however, this exaggeration in evoking the sensation results in overlooking the Advertisement message and failing in conveying it. For example, in Fig. 6 - top left image is supposed to convince the audience to buy the Five gum

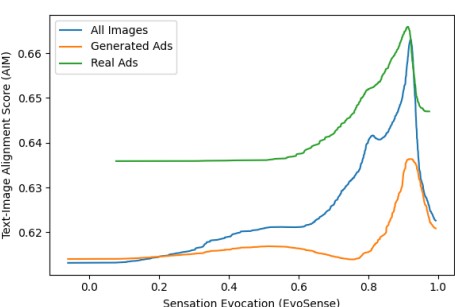

(a) Relationship between Sensation Evocation and Text-Image Alignment in real and generated advertisements.

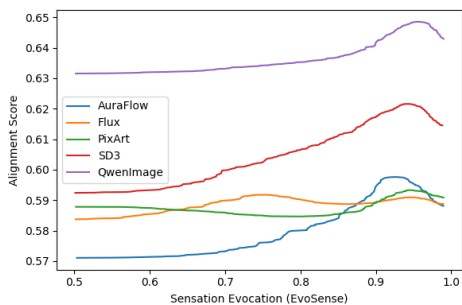

(b) Relationship between Sensation Evocation and Text-Image Alignment in generated advertisement by different T2I models in the benchmark.

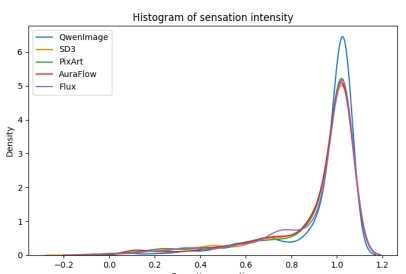

(c) Density of sensation intensity evoked by images generated by different T2I models.

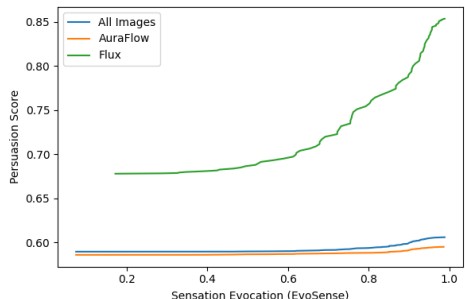

(d) Relation between sensation and persuasion.

Figure 7: Analysis of sensation intensity, text-image alignment, and persuasion in generated and real advertisements.

by showing the cooling and refreshing feature of the product; however, while the image intensely evokes the sensation it fails to convey the message.

We further examine the alignment between AR messages and images across varying sensation quality values. For alignment, we employ the AIM metric from Aghazadeh & Kovashka (2024), while sensation quality is measured using EvoSense. Although an overall trend exists in the relationship between alignment and sensation evocation, the plot exhibits considerable noise due to confounding factors such as image generation quality. To address this, we applied a Gaussian filter to the AIM scores to smooth the visualization. Fig. 7a presents the smoothed plot, which reveals that alignment initially increases with sensation evocation, reaches a peak, and subsequently decreases as sensation evocation continues to increase.

We then analyze the relation between sensation intensity and both alignment of the generated images by each T2I model in our benchmark in Fig. 7b. We observe, QwenImage while keeping the same behavior as other models, alignment increases with the increase of sensation intensity at first and decreases after, the alignment score of AIM metric is constantly higher which highlights the better performance of QwenImage in SensoryAd Generation. Finally, in Fig. 7c, show the density of sensation intensity values for images generated by each model. The plot in Fig. 7c further supports the exaggeration in evoking the sensation by generated images, by representing the highest density of sensation evocation value is in the domain that the alignment score decreases.

## A.2 CONNECTION OF PERSUASION AND SENSATION EVOCATION

While previously studied in marketing, and psychology area Lindstrom (2006); Yoon & Park (2012); Elder & Krishna (2022); Krishna et al. (2016) by doing the human study on sensory advertisement, in this section we analyze the relation between the persuasion and sensation evocation in generated images using computational persuasion evaluation metric from Aghazadeh & Kovashka (2024). We

plot the persuasion score over sensation intensity in the images in Fig. 7d, to analyze the effect of evoking sensation in making the images more persuasion. In the figure, we observe that the persuasion score for the images increases with the increase in the sensation intensity of the images.

**Ethical concerns around sensory advertisements.** There are two main implications: First generation of adversarial persuasive content such as encouraging the audience to drink alcohol more often. This is a general problem with any T2I model. Second, the model might generate sensitive content for a certain group of audience and this is one of the motivations for classification tasks. While automatically generating the Sensory Ads can be helpful, some sensitive sensations (for example pain) should be detected and prevented from showing to a specific group of audience. This is why it is also important to be able to classify the sensations evoked by the image.

### A.3 COMPARISON OF DIFFERENT SENSATIONS.

To analyze the capability of the T2I model in generating images evoking each sensation in our taxonomy, we isolated the sensation and only prompted the model to 'Generate an image that evokes {sensation}' with seeds from 0 to 9 resulting in 960 images for each model and 4800 images in overall. Fig. 8 represents the intensity of different sensations evoked in Sensory Image generation task. As shown in Fig. 8, models struggle more in evoking sensations with less common visual representation such as different human voice, or in overall different sounds. In contrast, models can evoke visual sensations - Sight and its children - with high intensity.

Fig. 9, shows an example of advertisements generated evoking four different sensations. Fig. 9, further represents the difference between capabilities of T2I models in evoking visual sensations like "Brilliance and Glow" and more abstract sensations like "Pressure".

### A.4 DATASET

To collect the dataset we defined the taxonomy represented in Fig. 10. Next, we randomly sampled 670 images from PittAd dataset Hussain et al. (2017) images covering 95 sensations and more than 40 topics. The diversity of images over 5 main sensations and 10 most frequent topics is represented in Fig. 11. For data annotation, we first had a test phase study on Prolific, gave the annotators detailed instruction with examples of images evoking each sensation, and selected a group of annotators based on the quality of their responses to do the main study. We used Qualtrics to create dynamic forms showing different options based on annotators choice in each step. The form is uploaded as the supplementary file.

The annotations were done by 12 annotators from different genders, within the age range of 25–60, and with education level of minimum high school diploma, achieving approval rate above 90% on more than 1000 annotations, and located in the United States. For each image, 1 annotator annotated the image, then the quality of annotations were approved by a skilled evaluator. If there was a disagreement on the annotation, the annotator was asked to explain the reasons for choosing a specific sensation (this happened very rarely), and if the second annotator was not convinced the annotation was ignored and the image was available in the pool for the new annotation. To further confirm the reliability of annotations for about 10% of images we collected 2 annotations from two different annotators and computed the $\kappa$ agreement between them. The human-human agreement in our dataset is 0.83 with 95% CI equal to [0.831, 0.838].

### A.5 EXPERIMENTAL SETUP DETAILS.

In this section we explain our experimental setup. We will also release the github link upon the acceptance of the paper. For all the models we used Hugging Face implementation of the models. **Sensation Classification.** In sensation classification tasks, we evaluated the model on real advertisement images in our dataset. We benchmarked MLLMs including the InternVL (InternVL3.5-8B), Gemma (gemma-3-4b-it), QwenVL (Qwen2.5-VL-7B-Instruct), and LLAVA-Next (llava-v1.6-vicuna-13b-hf) with 8-bit quantization for models with more than 4Billion parameters. We also benchmarked LLMs including Gemma, LLAMA3 (Meta-Llama-3-8B-Instruct), and QwenLM (Qwen2.5-7B-Instruct), given the descriptions generated by the same MLLMs. Similar to MLLMs 8-bit quantization was applied on models with more than 4B parameters.

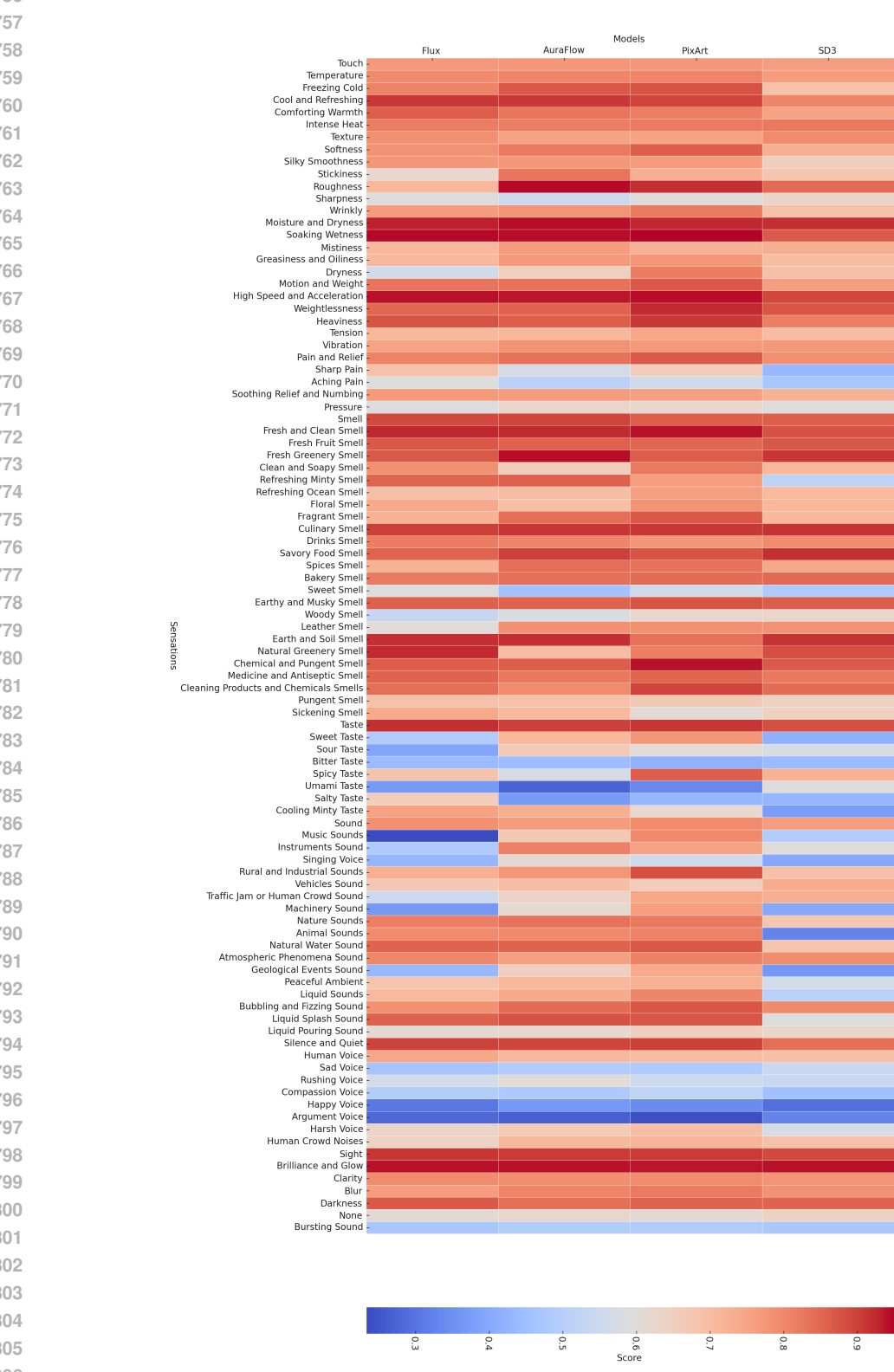

Figure 8: **Sensation Heatmap.** Average EvoSense score for images generated by each model for each sensation. Each model generates ten images evoking each sensation.

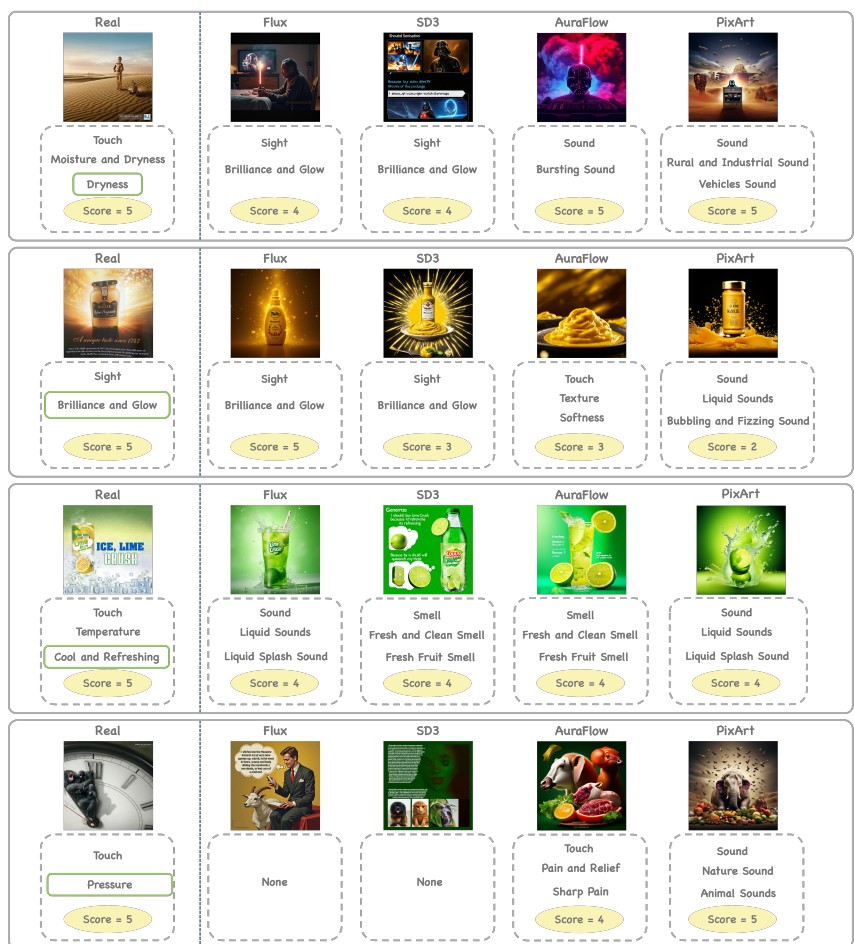

Figure 9: **Sensory Ad examples.** Four examples of real advertisement and generated advertisements by Flux (Black Forest Labs, 2024), SD3 (Esser et al., 2024), AuraFlow(Fal, 2024), and PixArt (Chen et al., 2024) given the action-reason interpretation and sensation annotation for the real advertisement. Green border represents the sensation used in the prompt of T2I models.

**EvoSense Training.** To train and evaluate our proposed evaluation metric, we randomly selected 50 images from annotations to create our training data. In our proposed training, we pair each two sensations with different intensity (scores chosen by human annotators) as chosen and rejected. Each data point in our training, included description of the image, chosen sensation, rejected sensation, and parent of chosen sensation. This training data setting resulted in 21000 data point. We fine-tuned the LLMs - LLAMA3 (Meta-Llama-3-8B-Instruct), and QwenLM (Qwen2.5-7B-Instruct) - using LoRA Hu et al. (2022) with batch-size of 1, and learning rate 5e-5. Our evaluation of EvoSense performance was on a subset of the images not selected for training.

**EvoSense Evaluation.** In this part we do a more in depth evaluation on EvoSense. We increase the number of images in our human-metric agreement evaluation to 100 images and more than 100000 (as before the images are unseen in the fine-tuning phase) and break-down the images by the high level sensation each image evokes and report the agreement under each category of sensation.

First, we do an ablation on number of fine-tuning steps (i.e. number of images in the fine-tuning set) to analyze the effect of fine-tuning data size on the performance of our evaluation metric. In table 4, we observe that the agreement of our metric with human annotators stays almost consistent with the increase in the size of fine-tuning data showing the effectiveness of our data expansion approach.

Next, we add InternVL, and QwenVL performing as a judge for sensation intensity to EvoSense baselines in our evaluation reporting the per sensation kappa agreement between human and metrics

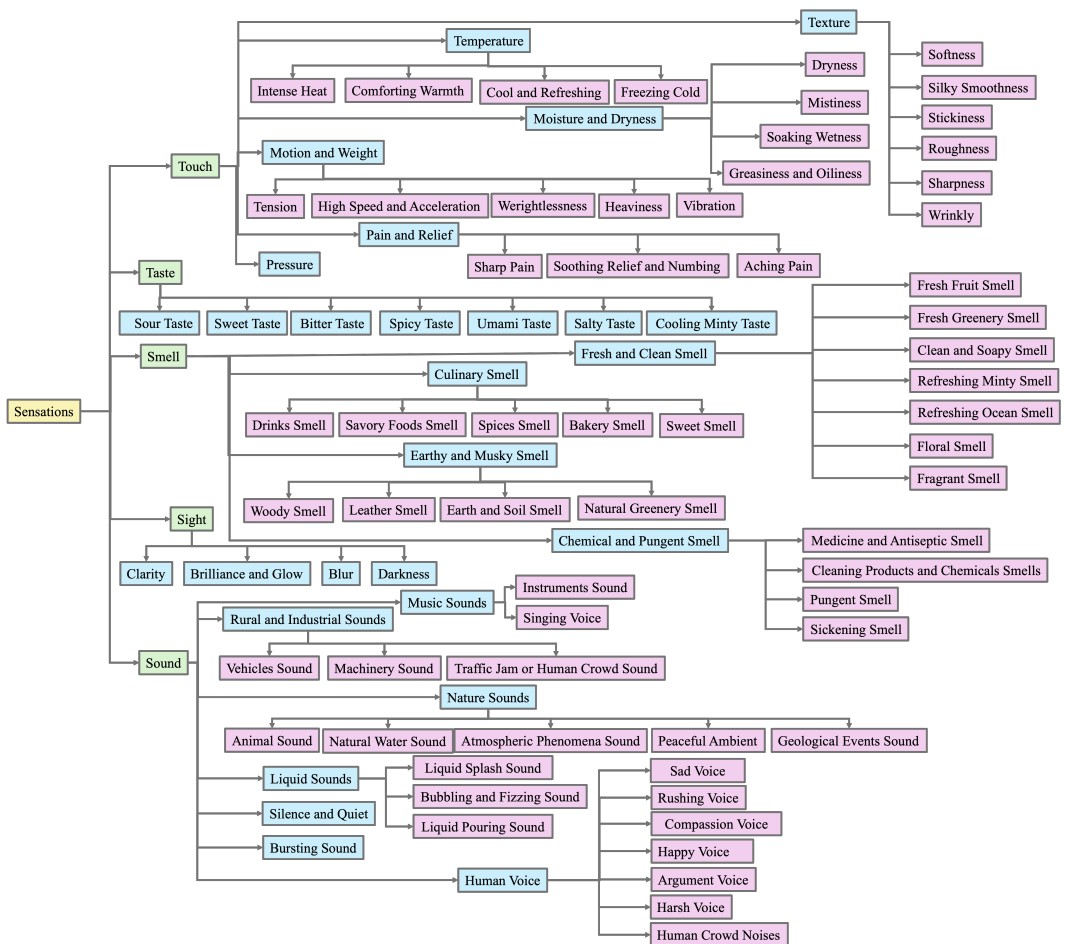

Figure 10: **Sensation Hierarchy.** First level, represents the main five human sensations, and each sensation is categorized into different set of sensations.

| Metrics | steps | touch | smell | sound | taste | sight | All |
|---|---|---|---|---|---|---|---|
| EvoSense (LLAMA3-instruct + $D_{InternVL}$) | 21000 | 0.79 | 0.82 | 0.77 | 0.84 | 0.85 | 0.80 |
| EvoSense (LLAMA3-instruct + $D_{InternVL}$) | 25000 | 0.80 | 0.82 | 0.78 | 0.83 | 0.88 | 0.81 |
| EvoSense (LLAMA3-instruct + $D_{InternVL}$) | 30000 | 0.80 | 0.82 | 0.78 | 0.84 | 0.88 | 0.81 |
| EvoSense (LLAMA3-instruct + $D_{InternVL}$) | 30000 | 0.80 | 0.81 | 0.78 | 0.84 | 0.87 | 0.81 |

Table 4: **Fine-tuning Ablation.** Kappa agreement ($\kappa$) between EvoSense metric with different number of fine-tuning steps on ¿10000 image-sensation pairs broken down into the sensation each image evokes among the high level sensations.

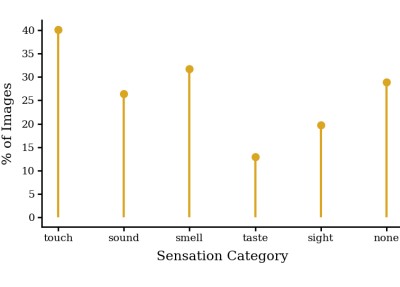

(a) Diversity of images over 5 main sensations.

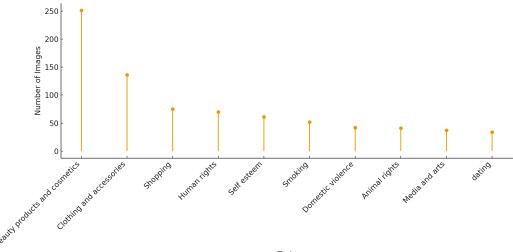

(b) Diversity of images over 10 most frequent topics in the SensoryAd dataset.

Figure 11: Image Distribution. Percentage of images evoking each high-level sensation category on left and distribution of different topics on right side.

| Metrics | touch | smell | sound | taste | sight | All | 95% CI |
|---|---|---|---|---|---|---|---|
| VQA-score | 0.58 | 0.60 | 0.42 | 0.65 | 0.58 | 0.57 | [0.561, 0.570] |
| Image-Reward | 0.49 | 0.50 | 0.38 | 0.34 | 0.45 | 0.46 | [0.457, 0.467] |
| CLIP-score | 0.48 | 0.47 | 0.36 | 0.41 | 0.30 | 0.44 | [0.430, 0.440] |
| Pick-score | 0.38 | 0.45 | 0.12 | 0.36 | 0.30 | 0.36 | [0.350, 0.361] |
| LLAMA3-instruct (zero-shot) | -0.09 | 0.08 | -0.22 | -0.005 | -0.007 | -0.04 | [-0.038, -0.027] |
| QwenLM (zero-shot) | -0.15 | 0.04 | -0.22 | 0.03 | 0.003 | -0.06 | [-0.064, -0.053] |
| InternVL (zero-shot) | 0.54 | 0.48 | 0.43 | 0.54 | 0.49 | 0.50 | [0.507, 0.514] |
| QwenVL (zero-shot) | 0.55 | 0.48 | 0.43 | 0.54 | 0.50 | 0.50 | [0.507, 0.514] |
| EvoSense (LLAMA3-instruct + $D_{InternVL}$) | **0.79** | **0.82** | **0.77** | **0.84** | **0.85** | **0.80** | [0.806, 0.813] |
| EvoSense (LLAMA3-instruct + $D_{QwenVL}$) | 0.76 | 0.77 | 0.70 | 0.79 | 0.73 | 0.75 | [0.754, 0.761] |
| EvoSense (QwenLM + $D_{InternVL}$) | 0.64 | 0.69 | 0.57 | 0.73 | 0.64 | 0.66 | [0.658, 0.666] |
| EvoSense (QwenLM + $D_{QwenVL}$) | 0.62 | 0.66 | 0.50 | 0.67 | 0.58 | 0.61 | [0.612, 0.621] |

Table 5: Kappa agreement between human annotators and evaluation metrics over different sensation categories on 100 images (10000 image-sensation pairs)

along with confidence intervals for all the image-sensation pairs. In table 5, it is observed that EvoSense improves the agreement with human by 60% compared to 0-shot MLLMs as a judge. Table 5 further represents that the agreement between EvoSense and human annotators is consistent over different sensation categories. EvoSense achieves higher agreement by at least 40% compared to baselines over different sensation categories.

**Kappa Agreement and Pearson Correlation Gap.** As observed in table 2, there is a big gap in the values of Kappa agreement ($\kappa$) and Pearson Correlation ($r$) reflected on all the metrics. In this part, we analyze the reason why the gap exist using a qualitative example of scores. The difference is because the annotators choose up-to 3 sensation groups evoked by the image, and the rest of the scores are 0. On the other hand, the computational metrics (including EvoSense and the baselines) choose different scores for each sensation. For computing $\kappa$ agreement, we use the sensation intensity as the criteria for choosing the winner sensation for the image given each two sensation. We ignore the sensation pairs where the human annotators assign the same score to both sensations. This way we significantly reduce the sparsity of human annotations for the image. So, while the incorrect sensations are included paired with selected sensations, they are not included as paired with other unselected sensations. This is why $\kappa$ is bigger than $r$ where the 0 scores are kept in correlation computation. Fig. 12a shows the scores from human and metrics for each sensation given the image highlighting the problem of correlation because of the sparsity of the human scores. The figure represents while high scores assigned by metric represent the sensations evoked by the image selected by the human, because of the sudden drop in the values of human scores, correlation becomes lower.

**Description Generation.** We generate descriptions of images with 0-shot InternVL, Gemma, and QwenVL and utilize the same descriptions in assessing LLMs' capabilities on sensation classifica-

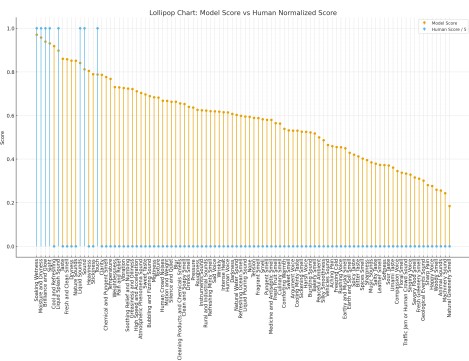

(a) Scores assigned to each sensation by human annotator and EvoSense.



(b) Example images.

Figure 12: Comparison of human scores and metric scores for each sensation's intensity evoked by the example image.

tion tasks, and EvoSense evaluation. Fig. 13, represents two examples of descriptions generated by each of the MLLMs. As shown in the examples, given the prompt in table 10 the models generate accurate descriptions of the image without interpreting the image. This prevents the information leakage in EvoSense while providing the accurate description of the image for LLMs in both classification and evaluation tasks. Negative agreements of zero-shot LLMs (LLAMA3-instruct and QwenLM) in table 2 further rejects the hypothesis of information leakage from MLLM description generation.

**SensoryAd Generation.** We benchmarked different T2I models including Stable Diffusion 3 (stable-diffusion-3-medium-diffusers), PixArt (PixArt-alpha/PixArt-XL-2-1024-MS), AuraFlow (AuraFlow-v0.3), Flux (FLUX.1-dev), and QwenImage (Qwen-Image) with 4-bit quantization on QwenImage and 8-bit quantization on rest of the models. We set the seed to 0 and number of time-steps as 28. For the rest of the model setting we use the default values. To generate the Sensory Ads, we utilized the sensation group (different level in hierarchy) evoked by the image with highest intensity generating an image for each sensation.

**Fine-tuning SD3 on SensoryAd Generation.** To further analyze the capability of T2I models, we fine-tuned the SD3 model on SensoryAd data. After fine-tuning text-image alignment of images increases by 0.01 compared to 0-shot SD3 and sensation intensity stays unchanged. We hypothesize, this is the result of implicitness of the text input and the sensation as previously suggested by Aghazadeh & Kovashka (2024) making the generation task more challenging.

**Prompts.** We have included prompts in Tables 10 (for image description generation), 9, 8, 7, 6 (for Sensation Classification tasks), and 12, 11 (for Sensory Image Generation tasks).

**Usage of AI:** We use AI to polish writing.

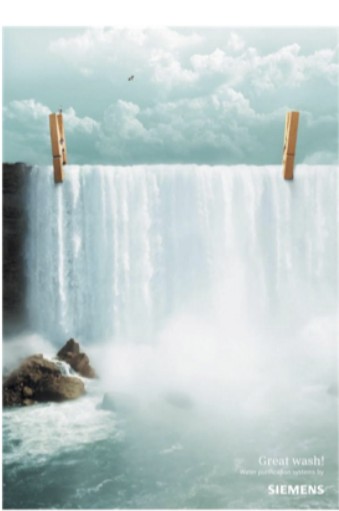

InternVL

Q1: Yes, there are objects in the image. The objects include: 1. A waterfall 2. Two wooden posts 3. A bird on the left wooden post 4. A bird in flight above the waterfall 5. Rocks at the base of the waterfall
Q2: The image features a large waterfall cascading down a cliff into a body of water below. Two wooden posts stand on either side of the waterfall, with a bird perched on the left post. Another bird is flying in the sky above the waterfall. The water at the base of the waterfall is turbulent, with visible rocks protruding from the water. The sky is partly cloudy, and the overall scene has a serene and natural appearance. In the bottom right corner, there is text that reads "Great wash! Water purification systems by SIEMENS."

Gemma

Q1: Waterfall, rocks, sky, clouds, clothespins, bird.
Q2: The image depicts a large waterfall cascading down a rocky cliff face into a body of water. The water is white and turbulent, creating a misty spray. Darker rocks are visible at the base of the waterfall. Above the waterfall, a blue sky with white, fluffy clouds dominates the upper portion of the image. Two wooden clothespins are attached to the waterfall at approximately the same height. A bird is flying across the sky. The bottom of the image shows the water flowing into a wider, calmer section of the body of water. The text "Great wash! Water purification systems by SIEMENS" is positioned at the bottom left corner of the image.

QwenVL

Q1: Yes, there are objects in the image, excluding text-only logos, and text. The objects are: – A bird flying in the sky – Two wooden clothespins
Q2: The image depicts a dramatic waterfall cascading over a rocky cliff into a body of water below. The water appears turbulent, creating mist as it falls. Two wooden clothespins are humorously placed at the top of the waterfall, one on each side, as if they are holding back the water. The sky above is partly cloudy with a soft blue hue. In the bottom right corner, there is text that reads "Great wash! Water purification systems by SIEMENS."

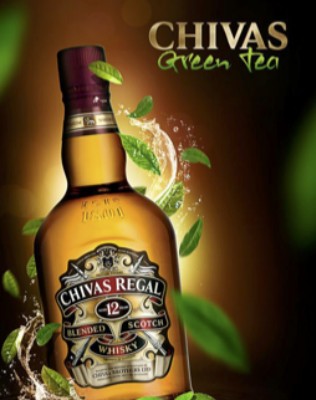

InternVL

Q1: Bottle, leaves, liquid splash, cap, label.
Q2: The image features a bottle of Chivas Regal 12 Years Old Blended Scotch Whisky. The bottle is prominently displayed against a warm, golden background. Green leaves are scattered around the bottle, with some leaves and a splash of liquid appearing to emerge from the bottle. The cap is purple, and the label on the bottle includes text and decorative elements. The text "CHIVAS Green Tea" is displayed at the top right of the image.

Gemma

"Q1: Bottle, green leaves, water splashes, dark background, and light reflections.
Q2: The image shows a brown bottle of Chivas Regal 12-year-old blended Scotch whisky. The bottle is positioned on a dark, slightly reflective surface. Several green leaves are scattered around the bottle, some with water splashes emanating from them. The background is dark, with hints of green and light reflections. The words "CHIVAS Green Tea" are displayed in a green, stylized font above the bottle. The bottle has a red-colored cap. There are also several more green leaves near the bottom of the image."

QwenVL

Q1: Yes, there are objects in the image, excluding text-only logos, and text. The objects are: – A bottle of Chivas Regal whisky – Green leaves – Water splashes
Q2: The image features a bottle of Chivas Regal 12-year-old Blended Scotch Whisky with a label prominently displayed. The bottle is set against a dark background with a warm, glowing light effect. Surrounding the bottle are green leaves and water splashes, creating a dynamic and refreshing visual. The text "CHIVAS Green Tea" is written in a stylized font, suggesting a fusion of the brand's signature whisky with green tea. The overall composition emphasizes the product's premium quality and the addition of green tea as a unique element.

Figure 13: **Image Description Examples.** Two examples of descriptions generated by InternVL, Gemma, and QwenVL. Both images are real advertisements from PittAdHussain et al. (2017) dataset.

Table 6: Prompt for LLM Hierarchical Sensation Classification

| Prompt |
| --- |
| **System**: You are a helpful assistant, choosing the sensations evoked by the described image given the following definition in ordered form. You can choose up to 3 sensations evoked by the image ranked in order of how well the sensations are evoked. If the image does not evoke any sensation you can choose None. 

 **Context**: 
 Sensation is the process of detecting and receiving information from the environment or the body through specialized sensory organs, which send signals to the brain for interpretation. 

 **Definition of the sensations in the options**: 
 {{context}} 

 **User**: What are the sensations evoked the most by the described image? Only return the indices of maximum of 3 options in ordered form without any further explanation. 

 **Image Description**: 
 {{description}} 

 **Options**: 
 {{options}} 

 Your answer must follow the following format: 
 Answer: ¡indices of maximum of 3 correct options separated by comma¿ |

Table 7: Prompt for MLLM Hierarchical Sensation Classification

| Prompt |
| --- |
| **System**: You are a helpful assistant, choosing the sensations evoked by the input image given the following definition in ordered form. You can choose up to 3 sensations evoked by the image ranked in order of how well the sensations are evoked. If the image does not evoke any sensation you can choose None.

**Context**:
Sensation is the process of detecting and receiving information from the environment or the body through specialized sensory organs, which send signals to the brain for interpretation.

**Definition of the sensations in the options**:
{{context}}

**User**: What are the sensations evoked the most by this image? Only return the indices of maximum of 3 options in ordered form without any further explanation.

**Options**:
{{options}}

Your answer must follow the following format:
Answer: ¡indices of maximum of 3 correct options separated by comma¿ |

Table 8: Prompt for LLM Multi-choice Sensation Classification

| Prompt |
| --- |
| **System**: You are a helpful assistant, choosing the sensations evoked by the described image given the following definition in ordered form. You are asked to choose all the sensations evoked by the image ranked in order of how well the sensations are evoked.If the image does not evoke any sensation you can choose None. |

**Context**:

Sensation is the process of detecting and receiving information from the environment or the body through specialized sensory organs, which send signals to the brain for interpretation.

**Definition of the sensations in the options**:
{{context}}

**User**: What are the sensations evoked the most by the described image? Only return the indices of the options in ordered form without any further explanation.

**Image Description**:
{{description}}

**Options**:
{{options}}

Your answer must follow the following format:
Answer: ¡indices of correct options separated by comma¿

Table 9: Prompt for MLLM Multi-choice Sensation Classification

| Prompt |
| --- |
| **System**: You are a helpful assistant, choosing the sensations evoked by the input image given the following definition in ordered form. You are asked to choose all the correct sensations evoked by the image ranked in order of how well the sensations are evoked. If the image does not evoke any sensation you can choose None. 

 **Context**: 
 Sensation is the process of detecting and receiving information from the environment or the body through specialized sensory organs, which send signals to the brain for interpretation. 

 **Definition of the sensations in the options**: 
 {{context}} 

 **User**: What are the sensations evoked the most by this image? Only return the indices of the options in ordered form without any further explanation. 

 **Options**: 
 {{options}} 

 Your answer must follow the following format: 
 Answer: ¡indices of correct options separated by comma¿ |

Table 10: Prompt for Structured Description Generation

| Prompt |
| --- |
| Carefully analyze the image and respond only in the specified format, without any interpretations or inferences. Focus on only the visible elements in the image. Ensure that any object seen in the image is included in Q1, even if it is described in more detail in Q2. 

 **Response Format:** 
 Q1: ${answer to Q1} 
 Q2: ${answer to Q2} 

 **Questions:** 
 Q1: Are there any objects in the image, excluding text-only logos, and text? List at most 5 such objects if present. 
 Q2: Describe the image in detail, focusing only on visible objects and elements without adding any interpretation, opinion, or analysis in a single paragraph. |

Table 11: Prompt for Sensory Image Generation

| Prompt |
| --- |
| Generate an image that evokes {{sensation}} sensation. |

Table 12: Prompt for Image Generation with Action-Reason and Sensation

| Prompt |
| --- |
| Generate an advertisement image that evokes {{sensation}} sensation and conveys the following messages: |
| {% for statement in action_reason %} |
| -{{statement}} |
| {% endfor %} |

