# OpenReview forum: "Sense it with your eyes: Sensation Generation and Understanding for Advertisements"
_ICLR.cc/2026/Conference — Submitted to ICLR 2026_

### Official Review · Reviewer_dfxP · 2025-10-26

**Soundness:** 2
**Presentation:** 3
**Contribution:** 2
**Rating:** 2
**Confidence:** 4

**Summary:**

This paper introduces the novel task of SensoryAd Generation, which aims to create advertisement images that not only convey a given message but also evoke a specified human sensation (e.g., temperature, pain). To support this, the authors construct the SensoryAd dataset, comprising real and generated ads annotated with a hierarchical taxonomy of sensations, the associated visual elements, and human ratings of evocation strength. They further propose a sensation classification task to benchmark AI models and introduce EvoSense, a novel evaluation method using a fine-tuned LLM to measure how effectively an image evokes a target sensation. Finally, the paper benchmarks current Text-to-Image (T2I) models, revealing their significant limitations in successfully completing the proposed SensoryAd generation task, often failing to balance sensation evocation with message clarity.

**Strengths:**

1. The idea is nice, to explore the different human sensation evoked by visual elements.

2. The writing is readable and easy to follow.

**Weaknesses:**

1. The paper doesn’t clearly describe how the annotation was conducted — for example, how many annotators participated, whether each sample was labeled by multiple people, or how disagreements were resolved. In addition, the annotators’ backgrounds (e.g., expertise, age range, gender, or cultural factors) can strongly influence subjective annotations, but these aspects are not discussed. Providing such information is essential for understanding the reliability of the dataset.

2. The dataset seems extremely small — only about 670 real samples and 75 generated ones. With such a small amount of data, it’s difficult to believe that the trained models can yield results with strong statistical or practical significance. The authors should either expand the dataset or discuss the limitations more explicitly.

3. It would be valuable if annotators not only provided a label but also a brief explanation of why a given visual element evoked a certain feeling. This could help establish a more interpretable and causally grounded link between the visual content and human sensation.

4. Since this task is highly related to human perception and preference, incorporating some reinforcement learning or reasoning-based mechanism to align the model with human judgments could make the generated advertisements more convincing and appealing. As it stands, the approach feels too passive and purely descriptive.

5. The dataset is small and coarse, the annotation lacks depth, and there’s no real algorithmic innovation or a well-defined end-to-end framework for advertisement image generation. The experiments also seem quite superficial, a few MLLMs are used as baselines, but none of the analyses go deep enough. The paper gives the impression of touching on many ideas without fully developing any of them.

6. The main text references materials in the appendix but doesn’t include clickable links or clear cross-references. As a result, readers have to manually search through the appendix to find the relevant content. This makes the paper unnecessarily hard to navigate.

7. The text mentions only a few hundred images, but Table 2 claims there are 5,000 real and 5,000 generated images. This inconsistency must be clarified — is it a typo, or were additional samples collected later?

8. Using MLLM-generated image descriptions as inputs for LLM-based sensation classification seems fundamentally flawed. Different MLLMs emphasize different aspects in their captions, and without human verification or standardization, these descriptions introduce large uncontrolled biases. Without any demonstrations or quality checks, evaluating LLMs based on such synthetic inputs is, in my view, methodologically unsound.

**Questions:**

See weakness.

---

> ### Author Response · Authors · 2025-11-26
> **Response 1/2**
>
> We thank the reviewer for the constructive feedback and the opportunity to respond and resolve the concerns. We are happy you found the idea nice to explore.
>
> ---
>
> ### [W1] Annotation Process
> > The paper doesn’t clearly describe how the annotation was conducted — for example, how many annotators participated, whether each sample was labeled by multiple people, or how disagreements were resolved. In addition, the annotators’ backgrounds (e.g., expertise, age range, gender, or cultural factors) can strongly influence subjective annotations, but these aspects are not discussed. Providing such information is essential for understanding the reliability of the dataset.
>
> The annotations were done by 12 annotators from different genders, within the age range of 25–60, and with education level of minimum high school diploma, achieving approval rate above 90% on more than 1000 annotations, and located in the United States. The 12 annotators were selected in the test phase of annotation where the annotators were provided with a detailed definition of the sensations and an example for each annotation and then were asked to annotate the images on sensation information.
> For each image, 1 annotator annotated the image, then the quality of annotations were approved by a skilled annotator (author). If there was a disagreement on the annotation, the annotator was asked to explain the reasons for choosing a specific sensation (this happened very rarely), and if the second annotator was not convinced the annotation was ignored and the image was available in the pool for the new annotation.
> * To further confirm the agreement and prevent the bias, for 60 images we use two annotators, and we compute the human - human agreement on 60 images (5760 image-sensation pairs) and the kappa agreement between the human annotators is 0.83 with 95% CI = [0.831, 0.838].
>
> ---
>
> ### [W2] Extension of Data Annotation
> > It would be valuable if annotators not only provided a label but also a brief explanation of why a given visual element evoked a certain feeling. This could help establish a more interpretable and causally grounded link between the visual content and human sensation.
>
> Thank you so much for the suggestion, we have and are currently collecting more data, and we are including the explanation annotation as well. We will release the data upon the acceptance of paper.
>
> ---
>
> ### [W3] RL with EvoSense Suggestion
> > Since this task is highly related to human perception and preference, incorporating some reinforcement learning or reasoning-based mechanism to align the model with human judgments could make the generated advertisements more convincing and appealing. As it stands, the approach feels too passive and purely descriptive.
>
> Thank you so much for your suggestion. As a paper submitted to benchmark track, in this work we are introducing the task, benchmark the models on the 3 proposed tasks, introduce the evaluation dataset, and propose the evaluation metric to evaluate the intensity of a sensation evoked by an image which is the significantly underexplored area in the field. We will use this suggestion in our future works. In this work, our goal was to discuss the gap of sensory images in current benchmarks on advertisement content.
>
> Also, we respectfully disagree that our metric is passive and purely descriptive. Given the gap in human-metric agreement between our metric and baseline metrics, a method to analyze the intensity of the sensation evoked by an image, especially with the current growth in the utilization of Generative AI in creating and understanding persuasive content.

---

> ### Author Response · Authors · 2025-11-26
> **Response 2/2**
>
> ### [W4] Paper's Contribution
> > The dataset is small and coarse, the annotation lacks depth, and there’s no real algorithmic innovation or a well-defined end-to-end framework for advertisement image generation. The experiments also seem quite superficial, a few MLLMs are used as baselines, but none of the analyses go deep enough. The paper gives the impression of touching on many ideas without fully developing any of them.
>
> * The goal of this paper as a paper submitted to the benchmark track, was addressing the underexplored tasks of understanding and generating sensory images, benchmarking the T2I models on the generation of Sensory Ads and LLMs/MLLMs on Sensation classification tasks. The goal of this paper was not the introduction of new methods for generation nor classification.
> * As our first contribution we introduce new tasks that have not been explored in the field while useful for media content generation and understanding. Our dataset is an evaluation dataset which considering previous evaluation datasets should not be a big concern. We also introduce an evaluation metric achieving high agreement with humans and showing improvement of +30% compared to baseline metrics.
> * **Size of Dataset**: Due to **annotation cost**, we only introduce the **evaluation dataset** for sensory advertisement. Compared to the recent evaluation dataset, our dataset scale is not small. Some of the recent similar publications with similar scale of dataset:
>   * Khanuja, Simran, et al. "An image speaks a thousand words, but can everyone listen? on image transcreation for cultural relevance."  **(Winner of EMNLP 2024 Best Paper Award) with 600 images in the introduced evaluation dataset**.
>   * Bitton-Guetta, Nitzan, et al. "Breaking Common Sense: WHOOPS! A Vision-and-Language Benchmark of Synthetic and Compositional Images" **(ICCV 2023) with 500 synthetic images in the introduced dataset.**
>
> * We would greatly appreciate it if you could also elaborate on the in-depth analysis that is expected for our paper, and it is currently missed. So, we can address and improve the quality of the paper.
>
> ---
>
> ### [W5] References to Appendix
> > The main text references materials in the appendix but doesn’t include clickable links or clear cross-references. As a result, readers have to manually search through the appendix to find the relevant content. This makes the paper unnecessarily hard to navigate.
>
> Thanks for the comment, we have addressed this in the updated paper.
>
> ---
>
> ### [W6] Data count
> > The text mentions only a few hundred images, but Table 2 claims there are 5,000 real and 5,000 generated images. This inconsistency must be clarified — is it a typo, or were additional samples collected later?
>
> Table 2 mentions 5000 image-sensation pairs. We evaluate each image for all 97 sensation labels (incorrect and correct) to make sure while the metric understands the correct sensation, it can also detect the incorrect sensations.
>
> ---
>
> ### [W7] MLLM Influence on LLM Performance
> > Using MLLM-generated image descriptions as inputs for LLM-based sensation classification seems fundamentally flawed. Different MLLMs emphasize different aspects in their captions, and without human verification or standardization, these descriptions introduce large uncontrolled biases. Without any demonstrations or quality checks, evaluating LLMs based on such synthetic inputs is, in my view, methodologically unsound.
>
> * We agree that we can hypothesize that part of the problem in LLMs when classifying the sensation can be the result of low quality verbalization. However, since our results on EvoSence shows that with the same descriptions the model can score the sensations with high agreement with human annotators showing the quality of descriptions. Moreover, the description is just describing the image and visual elements in the image.
> * We have added qualitative examples of MLLM generated descriptions to Appendix - A4 - Figure 13. In the examples, we show that the MLLMs provide a detailed and accurate description of images. Unfortunately, since we cannot attach images to the comments, we could not include the example in the comment as well.

---

### Official Review · Reviewer_S1yq · 2025-10-26

**Soundness:** 2
**Presentation:** 2
**Contribution:** 2
**Rating:** 2
**Confidence:** 4

**Summary:**

This paper proposes a new framework for understanding, evaluating and generating advertisement images that evoke human sensations, addressing the limitations of current Text-to-Image (T2I) models in this area. To accomplish this, the authors developed a hierarchical sensory taxonomy, which they used to construct the SensoryAd dataset, featuring detailed annotations on sensation type, intensity, and specific visual cues. They also introduce the SensoryAd Generation task, which requires generating an image that both conveys a given message and evokes a specified sensation. Critically, the paper proposes Evosense, a novel quantitative evaluation method that measures how effectively an image evokes a target sensation, utilizing a large language model (LLM) fine-tuned with a unique loss function that combines Contrastive Preference Optimization (CPO) with a Hierarchy Loss. Through several experiments, the results show that existing T2I models perform poorly on this specialized task. Meanwhile, EvoSense achieves a high agreement with human judgements, proving more reliable than other evaluation metrics like VQA-score.

**Strengths:**

# 1. Dataset and Evaluation Framework
This paper introduces a novel research direction that systematically explores the understanding and evaluation of advertisement images based on human sensations. It addresses the challenge of quantifying inherently abstract and subjective experiences by proposing a comprehensive framework. This framework includes a foundational dataset (SensoryAd) with detailed sensory annotations and an evaluation metric (EvoSense) designed to measure the intensity of evoked sensations.

# 2. Research Potential
The significance of this work lies in its systematic effort to model the intricate concept of human sensation within visual contexts. This contribution is expected to serve as a cornerstone for future research on generative models capable of understanding and reproducing implicit sensory experiences.

**Weaknesses:**

# 1. Validity of the Evaluation Metric
EvoSense first obtains an image description from an MLLM, then asks a text-only LLM to score the likelihood of the sensation term, reporting “average log-probability” as intensity. This setup may conflate lexical plausibility of captions with actual perceptual evocation in images. Moreover, the paper reports κ = 0.86 while the Pearson r is only 0.38 on real ads (Table 2), suggesting that categorical agreement might be inflated by frequent labels and does not convincingly validate the claimed intensity correlation.

# 2. Limitation of Generation
The SensoryAd Generation experiments rely entirely on textual prompting of existing models without any fine-tuning or conditioning on the SensoryAd dataset. Consequently, the results mostly reflect the models’ inherent prompt-following ability rather than learned sensation-aware generation.

# 3. Suitability of the Evaluation
The task requires conveying both the Action-Reason (AR) message and the target sensation. However, quantitative results emphasize only EvoSense intensity (Table 3). The paper itself notes that models often exaggerate the sensation and overlook the message (Fig. 6; Appendix A.1). Without a complementary message-alignment metric, it is difficult to balance or interpret the trade-off between sensation and message fidelity.

# 4. Sensation vs. Persuasion
The paper positions sensation evocation as a persuasive strategy in advertising and implicitly assumes that stronger sensations lead to greater persuasive effect. Yet no behavioral or attitudinal measures of persuasion (e.g., message recall, emotional engagement) are collected, and evaluation remains limited to sensation-intensity scores. Including a human study or a message-alignment metric could help substantiate the claimed link between sensory evocation and persuasive impact.

# 5. Annotation Reliability is Under-reported
Although the paper outlines a multi-stage annotation process on Prolific, it does not specify the number of annotators per image, the inter-annotator agreement for either labels or intensity scores, or how disagreements and outliers were handled. Since EvoSense training converts these ratings into pairwise preferences, the lack of clarity around annotation reliability raises questions about the stability of the supervision signal.

# 6. Small Annotated Subset of Generations
For each T2I model, only 15 out of 75 generated images were annotated, which limits the reliability of the model-level comparisons in Table 3. Moreover, using only 10 generated images reduces the interpretability of the sensation-wise heatmaps in Figure 7. The small sample size makes it difficult to assess performance on less visual sensations mentioned in the paper and may lead to sampling noise rather than consistent model differences.

# 7. Need for Clearer Motivation of Sensation Classification
The Sensation Classification task is presented as a key component but lacks clear justification within the overall framework. While it measures whether models can recognize sensory concepts, its connection to the later evaluation and generation tasks is under-explained, leaving it somewhat disconnected rather than foundational.


# 8. Typos and Grammatical Errors
There are multiple errors: “a an” in the abstract (p. 1); “the an” and “Hierarchal” instead of “Hierarchical” (p. 2); “fo” instead of “of” (p. 3); “Classfication” in A.3 (p. 15); “different between” in A.2 (p. 14); and “coveys” instead of “conveys” in Table 10 (p. 20).

**Questions:**

# 1. Scope of Sensation Across Advertisement Types
Given that the dataset includes both commercial and public service advertisements, could you clarify whether sensation plays an equally important role across these different advertisement categories, or if its importance varies depending on the ad type?


# 2. Relationship Between Classification Accuracy and EvoSense Agreement
Since both the classification and EvoSense tasks rely on the same annotated data, could you explain whether higher classification accuracy for a given sensation is associated with stronger agreement between EvoSense and human judgments across models or sensation categories?


# 3. Human Baseline for Sensation Classification
Table 1 reports model performance for MLLMs and LLMs, but no human baseline is included. Could you provide information on how consistent human annotators are when predicting sensations using the same taxonomy prompts, to help contextualize the reported F1 and R_parent scores?


# 4. Sampling Strategy for EvoSense Pairwise Training
In Appendix A.3, EvoSense is trained on 50 images expanded into about 21k pairwise instances. Could you describe how the positive and negative pairs were sampled, and whether the sampling was random across sensations or constrained within the same hierarchy branch?

---

> ### Author Response · Authors · 2025-11-26
> **Response 1/5**
>
> We thank the reviewer for the constructive feedback and the opportunity to respond and resolve the concerns. We are happy you found our research direction novel. We appreciate the acknowledgement of our systematic exploration on evaluation and understanding of sensory ads. We are thrilled that you found our work possibly serving as a cornerstone of future research.
>
> ---
>
> ### [W1] Validity of the Evaluation Metric
> > EvoSense first obtains an image description from an MLLM, then asks a text-only LLM to score the likelihood of the sensation term, reporting “average log-probability” as intensity. This setup may conflate lexical plausibility of captions with actual perceptual evocation in images.
>
> If the performance of EvoSense was the result of lexical plausibility of captions with actual perceptual evocation, the 0-shot LLMs in the same setting as EvoSense should have high agreement with human as well. However, we compare our metric **with the same setup but utilizing the zero-shot LLMs**. Since the **exact same descriptions** are given to both fine-tuned and zero-shot LLMs, the agreement of zero-shot version should be high as well given the hypothesis. However, **the zero-shot version results in negative agreement** highlighting the importance of the fine-tuning.
>
> * We have added qualitative examples of MLLM generated descriptions to appendix - A4 - Figure 13. In the examples, we show that the MLLMs only provide the detailed description for the image without further interpretations of the image itself or the sensation that it evokes. Unfortunately, since we cannot attach images to the comments, we could not include the example in the comment as well.
>
> * Finally, we have added 0-shot MLLMs (as a judge for sensation evocation) as the baseline for our metric showing that our evaluation method improves the agreement by 60% compared to MLLMs as a judge. This suggests that MLLMs, while capable of generating accurate descriptions, struggle to understand and find the correct sensation (as also shown in the table 1 in classification task), so, it is less probable for MLLMs to include the sensation in the description.
>
> |                   Metric                    | touch | smell | sound | taste | sight |   All    |
> |:-------------------------------------------:| :---: | :---: | :---: | :---: | :---: |:--------:|
> |                  VQAScore                   | 0.58 | 0.60 | 0.42 | 0.65 | 0.58 |   0.57   |
> |                  PickScore                  | 0.38 | 0.45 | 0.12 | 0.36 | 0.30 |   0.36   |
> |                  CLIPScore                  | 0.48 | 0.47 | 0.36 | 0.41 | 0.30 |   0.44   |
> |                Image-Reward                 | 0.49 | 0.50 | 0.38 | 0.34 | 0.45 |   0.46   |
> |      EvoSense (LLAMA3-instruct 0-shot)      | -0.09 | 0.08 | -0.22 | -0.005 | -0.007 |  -0.04   |
> |          EvoSense (QwenLM 0-shot)           | -0.15 | 0.04 | -0.22 | 0.03 | 0.003 |  -0.06   |
> |           **InternVL as a judge**           | 0.54 | 0.48 | 0.43 | 0.54 | 0.49 |   0.50   |
> |            **QwenVL as a judge**            | 0.55 | 0.48 | 0.43 | 0.54 | 0.50 |   0.50   |
> | EvoSense (LLAMA3-instruct + $D_{InternVL}$) | 0.79 | 0.82 | 0.77 | 0.84 | 0.85 | 0.80     |
> |  EvoSense (LLAMA3-instruct + $D_{QWenVL}$)  | 0.76 | 0.77 | 0.70 | 0.79 | 0.73 |   0.76   |
> |     EvoSense (QwenLM + $D_{InternVL}$)      | 0.64 | 0.69 | 0.57 | 0.73 | 0.64 |   0.66   |
> |      EvoSense (QwenLM + $D_{QWenVL}$)       | 0.62 | 0.66 | 0.50 | 0.67 | 0.58 |   0.61   |
>
> (We have bolded MLLMs as a judge)
>
> > Moreover, the paper reports κ = 0.86 while the Pearson r is only 0.38 on real ads (Table 2), suggesting that categorical agreement might be inflated by frequent labels and does not convincingly validate the claimed intensity correlation.
>
> * The difference **(as also reflected on baseline metrics)** is because the annotators choose up-to 3 sensation groups evoked by the image, and the rest of the scores are 0. On the other hand, the computational metrics (including EvoSense and the baselines) choose different scores for each sensation. For computing κ agreement, we use the sensation intensity as the criteria for choosing the winner sensation for the image given each two sensation. We ignore the sensation pairs where the human annotators assign the same score to both sensations.
> This way we significantly reduce the sparsity of human annotations for the image. So, while the incorrect sensations are included paired with selected sensations, they are not included as paired with other unselected sensations. This is why κ is bigger than r.
> * We have added an example sensation score for an image to the Appendix - A5 - Figure 12, showing the problem of correlation because of the **sparsity of the human scores**. The figure represents while high scores assigned by metric represent the sensations evoked by the image selected by the human, because of the sudden drop in the values of human scores, correlation becomes lower.

---

> ### Author Response · Authors · 2025-11-26
> **Response 2/5**
>
> ### [W2] Limitation of Generation
> >The SensoryAd Generation experiments rely entirely on textual prompting of existing models without any fine-tuning or conditioning on the SensoryAd dataset. Consequently, the results mostly reflect the models’ inherent prompt-following ability rather than learned sensation-aware generation.
>
> Thank you so much for the insightful suggestion of fine-tuning. The goal of this benchmark is to show the need for more sensory image data or further improvement in the generalizability of models. We have fine-tuned the SD3 on the sensory ad data and the AIM (Alignment of Image and Message) was increased from 0.61 to 0.62 while EvoSense stayed on unchanged 0.89 for fine-tuned SD3. We've included this ablation discussion to the Appendix - A5 section.
>
> ---
>
> ### [W3] Suitability of the Evaluation
> > The task requires conveying both the Action-Reason (AR) message and the target sensation. However, quantitative results emphasize only EvoSense intensity (Table 3).
>
> Thank you so much for the constructive comment.
> We have added the text-image alignment (AIM from [1]) column to table 2 in the paper.
> We observe that except for QwenImage **higher values for sensation evocation results in lower alignment score**. For QwenImage, while the sensation score is higher, it has higher alignment with messages.
>
> To further analyze the difference in QwenImage behaviour we have added the plot representing the sensation intensity (x-axis) and alignment score (y-axis) to Appendix - A1 (Fig. 7 - b). We observe **while keeping the same behaviour as other models** - alignment increases with the increase of sensation intensity at first and decreases after - the **alignment score of AIM metric is constantly higher** which highlights the better performance of QwenImage in SensoryAd Generation.
> We also show the density of sensation intensity values for images generated by each model in Appendix - A1 (Fig. 7 - c) and in the plot we show that similar to other models, QwenImage exaggerates in evoking the sensation (the highest density of sensation evocation value is in the domain that the alignment score decreases).
>
> [1] [Aghazadeh, Aysan, and Adriana Kovashka. "Cap: Evaluation of persuasive and creative image generation." Proceedings of the IEEE/CVF International Conference on Computer Vision. 2025.](https://openaccess.thecvf.com/content/ICCV2025/html/Aghazadeh_CAP_Evaluation_of_Persuasive_and_Creative_Image_Generation_ICCV_2025_paper.html)
>
> > The paper itself notes that models often exaggerate the sensation and overlook the message (Fig. 6; Appendix A.1). Without a complementary message-alignment metric, it is difficult to balance or interpret the trade-off between sensation and message fidelity.
>
> We have added the plot (Fig. 7 - a) showing the relation between the alignment and sensation evocation to appendix A.1. The plot suggests that at first by increasing the sensation evocation text-image alignment increases and then it decreases with increase of sensation quality. This supports our claim that exaggeration exists.

---

> ### Author Response · Authors · 2025-11-26
> **Response 3/5**
>
> ---
>
> ### [W4] Sensation vs. Persuasion
> > The paper positions sensation evocation as a persuasive strategy in advertising and implicitly assumes that stronger sensations lead to greater persuasive effect. Yet no behavioral or attitudinal measures of persuasion (e.g., message recall, emotional engagement) are collected, and evaluation remains limited to sensation-intensity scores. Including a human study or a message-alignment metric could help substantiate the claimed link between sensory evocation and persuasive impact.
>
> * **Marketing Publications**
>
>     We are not discovering the relation between the sensation and persuasion, instead we are building on prior marketing and psychology publications to explore the machine learning techniques in this area. As also referenced in the paper, Sensory Advertisement is a well-studied concept in marketing research. Since this has been previously extensively studied we rely on and trust the outcome of previous marketing and psychology works. The claim is already supported by the following papers, and books from marketing research including the user study.
>
>     Lindstrom in his book [2] cited over 1400 times, shows the relation between the sensory appeal and brand effectiveness, and shows that the brands should use senses based on their products. They further suggest that the sensory perception in marketing results in consumers’ loyalty to the brand.
>
>     [3] cited over 130 times, designs two studies on relation between sensory advertisements and attitude toward the brand. They show sensory ads increase positive attitude toward the brand (increasing the effectiveness of advertisement) and the reason for why sensory advertisement works is that it triggers the self-referencing, i.e.  audience imagining using the product.
>
>     [4] cited over 380 times, discusses how Sensory Marketing influences the effectiveness of the advertisement highlighting improvement in the effectiveness of ads by evoking mental simulation.
>
>     [5] cited over 130 times, reviewed the existing works in sensory images and their effects on consumer behavior, and their findings suggest that sensory advertisements increase the mental imagery in consumers and mental imagery can enhance the ad persuasion.
>
>     [2] [Lindstrom, Martin. "Brand sense: How to build powerful brands through touch, taste, smell, sight and sound." Strategic Direction 22.2 (2006).](https://www.emerald.com/sd/article/doi/10.1108/sd.2006.05622bae.001/353448/Brand-Sense-How-to-Build-Powerful-Brands-Through)
>
>     [3] [Yoon, Sung-Joon, and Ji Eun Park. "Do sensory ad appeals influence brand attitude?." Journal of Business Research65.11 (2012): 1534-1542.](https://www.sciencedirect.com/science/article/pii/S0148296311000701)
>
>     [4] [Krishna, Aradhna, Luca Cian, and Tatiana Sokolova. "The power of sensory marketing in advertising." Current Opinion in Psychology 10 (2016): 142-147.](https://www.sciencedirect.com/science/article/abs/pii/S2352250X16000099)
>
>     [5] [Elder, Ryan S., and Aradhna Krishna. "A review of sensory imagery for consumer psychology." Journal of Consumer Psychology 32.2 (2022): 293-315.](https://myscp.onlinelibrary.wiley.com/doi/full/10.1002/jcpy.1242)
>
> * **Persuasion over Sensation Plot**
>
>     We have also plotted the persuasion score computed using Persuasion metric from [1] over the sensation scores for different generated images. We have added the plot to the Appendix - A1 - Figure 7-d. We observe that the persuasion increases with the increase of sensation intensity.
>
>     [1] [Aghazadeh, Aysan, and Adriana Kovashka. "Cap: Evaluation of persuasive and creative image generation." Proceedings of the IEEE/CVF International Conference on Computer Vision. 2025.](https://openaccess.thecvf.com/content/ICCV2025/html/Aghazadeh_CAP_Evaluation_of_Persuasive_and_Creative_Image_Generation_ICCV_2025_paper.html)

---

> ### Author Response · Authors · 2025-11-26
> **Response 4/5**
>
> ### [W5] Annotation Reliability is Under-reported
> > Although the paper outlines a multi-stage annotation process on Prolific, it does not specify the number of annotators per image, the inter-annotator agreement for either labels or intensity scores, or how disagreements and outliers were handled. Since EvoSense training converts these ratings into pairwise preferences, the lack of clarity around annotation reliability raises questions about the stability of the supervision signal.
>
> * **Annotation Process**
>
>     The annotations were done by 12 annotators from different genders, within the age range of 25–60, and with education level of minimum high school diploma, achieving approval rate above 90% on more than 1000 annotations, and located in the United States. The 12 annotators were selected in the test phase of annotation where the annotators were provided with a detailed definition of the sensations and an example for each annotation and then were asked to annotate the images on sensation information.
> For each image, 1 annotator annotated the image, then the quality of annotations were approved by a skilled evaluator. If there was a disagreement on the annotation, the annotator was asked to explain the reasons for choosing a specific sensation (this happened very rarely), and if the second annotator was not convinced the annotation was ignored and the image was available in the pool for the new annotation. We have updated and added these information to Appendix - A4 section.
>
> * **Human-Human Agreement**
>
>     To further confirm the agreement and prevent the bias we compute the human - human agreement on 60 images (5760 image-sensation pairs) and the kappa agreement between the human annotators is 0.83 with 95% CI = [0.831, 0.838].
>
> ---
>
> ### [W6] Small Annotated Subset of Generations
> > For each T2I model, only 15 out of 75 generated images were annotated, which limits the reliability of the model-level comparisons in Table 3.
>
> We appreciate the reviewer’s concern and would like to clarify the role of annotated vs. generated images in our evaluation.
> Our benchmark produces 1,660 images per model (700 Sensory Ad + 960 Sensory Image), totaling 8,300 generated images across all models. These are the images used in Table 3 to compare model-level sensation evocation performance. Importantly, Table 3 is not based on the 75 manually annotated images, but instead uses EvoSense scores applied to the full generated set, which ensures stable estimates at the model level.
> * The 75 annotated images are used only in Table 2 to validate EvoSense against human judgments. We have now added confidence intervals to further quantify uncertainty.
>
> > Moreover, using only 10 generated images reduces the interpretability of the sensation-wise heatmaps in Figure 7. The small sample size makes it difficult to assess performance on less visual sensations mentioned in the paper and may lead to sampling noise rather than consistent model differences.
>
> Figure 7 is an illustrative heatmap intended to visualize relative patterns in sensation difficulty, not to provide statistical estimates. Although it uses numeric values, the underlying sample size (10 images per model for each sensation, and 40 per sensation) is not intended for model comparison.
>
> ---
>
> ### [W7] Need for Clearer Motivation of Sensation Classification
> > The Sensation Classification task is presented as a key component but lacks clear justification within the overall framework. While it measures whether models can recognize sensory concepts, its connection to the later evaluation and generation tasks is under-explained, leaving it somewhat disconnected rather than foundational.
>
> Thank you so much for the insightful comment. We have added the motivation for the classification task to the paper. Here are the two main motivations:
> * Some sensations like pain sensation can be sensitive to a group of audience such as children in a certain age. Given this, to prevent the presentation of a specific sensation to a specific group, the filtering systems should be able to detect the sensations evoked by the content.
> * To evaluate if the images evoke specific sensation and how well the image evokes the sensation, it first needs to find the correct sensation evoked by the image. In fact, we use the results from table 1 as a motivation for fine-tuning the LLM in EvoSense. The low performance of models in classifying the sensation shows that these models cannot perform as a judge for sensation evocation.
>
> ---
>
> ### [W8] Typos and Grammatical Errors
> > There are multiple errors: “a an” in the abstract (p. 1); “the an” and “Hierarchal” instead of “Hierarchical” (p. 2); “fo” instead of “of” (p. 3); “Classfication” in A.3 (p. 15); “different between” in A.2 (p. 14); and “coveys” instead of “conveys” in Table 10 (p. 20).
>
> We have addressed these in the updated paper.

---

> ### Author Response · Authors · 2025-11-26
> **Response 5/5**
>
> ### [Q1] Relation between Classification Task Performance and EvoSense Performance
> > Since both the classification and EvoSense tasks rely on the same annotated data, could you explain whether higher classification accuracy for a given sensation is associated with stronger agreement between EvoSense and human judgments across models or sensation categories?
>
> To answer this question, we break down the agreement evaluation into 5 main sensation categories and add 0-shot MLLMs as the baseline for our evaluation metric. We show the agreement between human annotators and metrics on 100 images in the following table:
>
> |                   Metric                    | touch | smell | sound | taste | sight |   All    |
> |:-------------------------------------------:| :---: | :---: | :---: | :---: | :---: |:--------:|
> |                  VQAScore                   | 0.58 | 0.60 | 0.42 | 0.65 | 0.58 |   0.57   |
> |                  PickScore                  | 0.38 | 0.45 | 0.12 | 0.36 | 0.30 |   0.36   |
> |                  CLIPScore                  | 0.48 | 0.47 | 0.36 | 0.41 | 0.30 |   0.44   |
> |                Image-Reward                 | 0.49 | 0.50 | 0.38 | 0.34 | 0.45 |   0.46   |
> |      EvoSense (LLAMA3-instruct 0-shot)      | -0.09 | 0.08 | -0.22 | -0.005 | -0.007 |  -0.04   |
> |          EvoSense (QwenLM 0-shot)           | -0.15 | 0.04 | -0.22 | 0.03 | 0.003 |  -0.06   |
> |           **InternVL as a judge**           | 0.54 | 0.48 | 0.43 | 0.54 | 0.49 |   0.50   |
> |            **QwenVL as a judge**            | 0.55 | 0.48 | 0.43 | 0.54 | 0.50 |   0.50   |
> | EvoSense (LLAMA3-instruct + $D_{InternVL}$) | 0.79 | 0.82 | 0.77 | 0.84 | 0.85 | 0.80     |
> |  EvoSense (LLAMA3-instruct + $D_{QWenVL}$)  | 0.76 | 0.77 | 0.70 | 0.79 | 0.73 |   0.76   |
> |     EvoSense (QwenLM + $D_{InternVL}$)      | 0.64 | 0.69 | 0.57 | 0.73 | 0.64 |   0.66   |
> |      EvoSense (QwenLM + $D_{QWenVL}$)       | 0.62 | 0.66 | 0.50 | 0.67 | 0.58 |   0.61   |
>
>
> * 0-shot MLLMs also utilized in classification tasks perform better on touch sensation category compared to smell while this opposite in EvoSense metric.
> The same happens between taste and sight sensations. While MLLMs perform better on taste, EvoSense with LLAMA3-instruct on descriptions generated by InternVL, perform higher sight.
>   * This suggests that such association between MLLM performance and EvoSense does not exist.
>
> ---
>
> ### [Q2] Human-Human Agreement on Sensation Annotation
> > Table 1 reports model performance for MLLMs and LLMs, but no human baseline is included. Could you provide information on how consistent human annotators are when predicting sensations using the same taxonomy prompts, to help contextualize the reported F1 and R_parent scores?
>
> We compute the human - human agreement on 60 images (5760 image-sensation pairs) and the kappa agreement between the human annotators is 0.83 with 95% CI = [0.831, 0.838].
>
> ---
> ### [Q3] Fine-tuning Data
> >In Appendix A.3, EvoSense is trained on 50 images expanded into about 21k pairwise instances. Could you describe how the positive and negative pairs were sampled, and whether the sampling was random across sensations or constrained within the same hierarchy branch?
>
> * The images in fine-tuning are chosen randomly resulting in random selection of the positive sensations in the sample set.
> * The positive and negative pairs are chosen given the sensation scores from the annotation. For each image there are a maximum of 3 sensation groups (parent1, parent 2, …, child sensation). Each of these sensation groups are annotated with scores from 1 to 5. The score for the rest of the sensations is 0. Each two sensations are compared to each other and if the scores are not equal the sensation with higher score is added as the positive sensation and the lower one is the negative sensation. So, each sensation can be negative when paired with higher score sensation and positive when paired with lower score sensation.

---

### Official Review · Reviewer_FpSs · 2025-11-02

**Soundness:** 2
**Presentation:** 2
**Contribution:** 2
**Rating:** 2
**Confidence:** 4

**Summary:**

The paper introduces SensoryAd, a benchmark and dataset for studying how advertisements evoke human sensations such as touch, taste, or temperature through visual cues. It defines two classification tasks for identifying sensations in ad images and a generation task aimed at producing ads that evoke specified sensations. To evaluate these, the authors propose EvoSense, a two-stage evaluation metric that uses a multimodal large language model to describe images and a fine-tuned language model (with contrastive and hierarchical losses) to estimate the intensity of evoked sensations. Experiments on 670 annotated ad images show that EvoSense correlates strongly with human judgments and reveals that current text-to-image models often exaggerate sensations while losing the intended advertising message.

**Strengths:**

1. The paper introduces a genuinely novel research angle: studying how visual advertisements evoke sensations beyond emotion or aesthetics. This focus on multi-sensory evocation represents a fresh and underexplored problem in multimodal learning.

2. The benchmark design is cohesive. The paper proposes a complete setup that includes a hierarchical taxonomy, well-defined sensation-classification tasks, and a generation task. Together, these form a usable foundation for further research on sensory perception in vision-language models.

3. The qualitative analyses are insightful. The examples showing “sensory exaggeration” highlight an important real-world failure mode of text-to-image models—overemphasizing sensory cues while neglecting message fidelity. This kind of observation can inform future model design and evaluation.

4. The paper’s overall presentation is clear and well-organized. The authors articulate the problem, dataset construction, and evaluation pipeline in a logical sequence, making the contributions understandable even to readers outside of advertising research.

**Weaknesses:**

1. The paper claims to develop a taxonomy of senses at different levels of granularity, where the first layer corresponds to the five fundamental sensory modalities.
   a. The description of annotation guidelines is vague. The statement that “each annotator was tested and approved based on completing a practice form after reading detailed instructions, definitions of sensations, and illustrative examples” is too high-level to assess reliability. It is unclear what exact criteria or examples were used, how consistency was ensured, or how ambiguous sensations were handled.
   b. It is also unclear how this sensory taxonomy meaningfully differs from the original PittAd dataset, which already contains message-based annotations (“I should do X because Y”) that capture intent and affect. The added sensory layer may be overlapping conceptually without sufficient justification for being treated as a distinct modality space.

2. The paper lacks comparisons with strong zero-shot and few-shot baselines such as GPT or large instruction-tuned multimodal models, as well as simple, transparent baselines like those used in Behavior-LLaVA. Without these comparisons, it is difficult to judge whether EvoSense or the proposed classification approach truly adds value beyond general-purpose reasoning models.

3. Dataset scale and diversity remain limited. The core corpus of 670 PittAd images (plus a small generated subset) is too small for training robust perception or evaluation models and risks domain bias. The reported 21k datapoints are augmented variations of this same source, limiting diversity rather than expanding it. Furthermore, there is a risk of evaluation circularity: EvoSense is fine-tuned on the same annotation distribution that it later evaluates. Even with held-out images, the taxonomy, task definitions, and metric are all co-designed, leaving open the possibility of domain overfitting or leakage from description-generation stages.

4. The claim that this is “the first investigation of how ads evoke the senses” is inaccurate. Prior work such as Behavior-LLaVA (ICLR) and related multimodal behavior modeling studies have already explored the link between generated content and human reactions, across a broader behavioral space—including Reddit upvotes, YouTube likes, replay rates, saliency, memorability, and comment patterns. Those efforts provide a more comprehensive foundation for connecting visual content to audience response.

5. A major conceptual weakness is that the paper equates sensation intensity with persuasive effectiveness without providing empirical behavioral validation. While the authors frame sensory evocation as “a persuasive strategy” and “a crucial element of persuasive advertising,” they offer no A/B or user studies (e.g., click-throughs, recall, purchase intent) to substantiate this claim. The link between sensory evocation and actual persuasion therefore remains hypothetical rather than experimentally demonstrated.

6. The T2I benchmarking is shallow. Only five models are tested, with approximately 75 images per model, fixed random seeds, and some quantization. This limited scope is insufficient to draw generalizable conclusions across a rapidly evolving T2I ecosystem, especially given the variation in prompt formats, diffusion steps, and stylistic priors across models.

7. The paper reports a κ = 0.86 “near-perfect” agreement between EvoSense and human annotators but does not provide confidence intervals, standard errors, or per-sensation breakdowns, which limits interpretability. Comparing κ across model types is statistically problematic because κ depends on class prevalence, which differs between LLM-only and MLLM-based setups. There is also no power or uncertainty analysis to demonstrate that these results are stable given the small dataset. All sensations are aggregated into a single κ/r score, even though some (e.g., “Glow”) are visually trivial while others (e.g., “Dryness”) are abstract and harder to ground. Without stratified or normalized results, it is unclear whether the model’s high κ reflects true generalization or dominance by a few easy classes. Consequently, the claim of “human-level” agreement is numerically strong but statistically under-supported.

**Questions:**

NA

---

> ### Author Response · Authors · 2025-11-26
> **Response 1/5**
>
> We thank the reviewer for the constructive feedback and the opportunity to respond and resolve the concerns. We are happy they found our research angle genuinely novel and the qualitative analysis insightful. We are glad they find the benchmarks cohesive and useful for creating the foundation for future research.
>
> ---
>
> ### [W1] EvoSense Baselines
> > The paper lacks comparisons with strong zero-shot and few-shot baselines such as GPT or large instruction-tuned multimodal models, as well as simple, transparent baselines like those used in Behavior-LLaVA. Without these comparisons, it is difficult to judge whether EvoSense or the proposed classification approach truly adds value beyond general-purpose reasoning models.
>
> * EvoSense is not a classification method, and this paper is not proposing a classification method. In this work, we propose the classification tasks, and an evaluation metric (EvoSense) to evaluate how well an image evokes the sensation.
> We have compared our evaluation metrics with strong zero-shot models of LLAMA3-instruct and QwenLM both instruction-tuned. The results on state-of-the-art zero-shot models show negative agreement with humans.
> * For the classification tasks, we use state-of-the-art MLLMs (all instruction tuned) and show they fail in correctly classifying and understanding the sensations evoked by the images.
> * We have added the MLLMs used for description generation for EvoSense (both instruction-tuned) as a judge for sensation evocation. As shown in the table our method achieves 60% higher agreement with humans compared to MLLMs as a judge. We show the Kappa agreement of different baselines (+ the MLLMs judges) with human annotators on 100 images (10000 image-sensation pairs) in the following table:
>
> |                   Metric                    | touch | smell | sound | taste | sight |   All    |
> |:-------------------------------------------:| :---: | :---: | :---: | :---: | :---: |:--------:|
> |                  VQAScore                   | 0.58 | 0.60 | 0.42 | 0.65 | 0.58 |   0.57   |
> |                  PickScore                  | 0.38 | 0.45 | 0.12 | 0.36 | 0.30 |   0.36   |
> |                  CLIPScore                  | 0.48 | 0.47 | 0.36 | 0.41 | 0.30 |   0.44   |
> |                Image-Reward                 | 0.49 | 0.50 | 0.38 | 0.34 | 0.45 |   0.46   |
> |      EvoSense (LLAMA3-instruct 0-shot)      | -0.09 | 0.08 | -0.22 | -0.005 | -0.007 |  -0.04   |
> |          EvoSense (QwenLM 0-shot)           | -0.15 | 0.04 | -0.22 | 0.03 | 0.003 |  -0.06   |
> |             **InternVL as a judge**             | 0.54 | 0.48 | 0.43 | 0.54 | 0.49 |   0.50   |
> |              **QwenVL as a judge**              | 0.55 | 0.48 | 0.43 | 0.54 | 0.50 |   0.50   |
> | EvoSense (LLAMA3-instruct + $D_{InternVL}$) | 0.79 | 0.82 | 0.77 | 0.84 | 0.85 | 0.80     |
> |  EvoSense (LLAMA3-instruct + $D_{QWenVL}$)   | 0.76 | 0.77 | 0.70 | 0.79 | 0.73 |   0.76   |
> |     EvoSense (QwenLM + $D_{InternVL}$)      | 0.64 | 0.69 | 0.57 | 0.73 | 0.64 |   0.66   |
> |      EvoSense (QwenLM + $D_{QWenVL}$)       | 0.62 | 0.66 | 0.50 | 0.67 | 0.58 |   0.61   |
>
> (We have bolded MLLMs as a judge)
>
> Our EvoSense metric improves the agreement with human compared to **0-shot instruction-tuned MLLMs by 60%**.

---

> ### Author Response · Authors · 2025-11-26
> **Response 2/5**
>
> ### [W2] Dataset Scale
>
> > Dataset scale and diversity remain limited. The core corpus of 670 PittAd images (plus a small generated subset) is too small for training robust perception or evaluation models and risks domain bias.
>
> Due to **annotation cost**, we only introduce the **evaluation dataset** for sensory advertisement. Compared to the recent evaluation dataset, our dataset scale is not small. Some of the recent similar publications with similar scale of dataset:
>   * Khanuja, Simran, et al. "An image speaks a thousand words, but can everyone listen? on image transcreation for cultural relevance."  **(Winner of EMNLP 2024 Best Paper Award) with 600 images in the introduced evaluation dataset**.
>   * Bitton-Guetta, Nitzan, et al. "Breaking Common Sense: WHOOPS! A Vision-and-Language Benchmark of Synthetic and Compositional Images" **(ICCV 2023) with 500 synthetic images in the introduced dataset.**
>
> > The reported 21k datapoints are augmented variations of this same source, limiting diversity rather than expanding it. Furthermore, there is a risk of evaluation circularity: EvoSense is fine-tuned on the same annotation distribution that it later evaluates.
>
> The images in the **train and test set are not the same.**
>
> For the train we use 50 images (now we also fine-tune on 100 images in new experiments), and use the annotations to use each image multiple times.
> **We do not use the augmentation method;** we only use the different annotations within our data. We explain the fine-tuning data creation utilizing each images' annotations:
> * Each image in our fine-tuning setting is paired with negative and positive sensation, which expands the data to 21000 samples that the models have been fine-tuned on.
> * Each image is annotated with a maximum of 9 sensations that the image evokes.
> * Each sensation is associated with a score on the quality of the sensation evocation.
> * For each image we collect the (positive, negative) sensation pairs for each two sensations:
>   1. If the score of one is higher than the other one then the sensation with higher score is positive and the one with lower score is negative sensation.
>   2. If the score of two sensations are equal then the pair is not included in the fine-tuning data.
> * This way each sensation might be positive in one pair and negative in another.
> * Given that there are 96 sensations in our taxonomy, and maximum of 9 sensations are only selected to be evoked the minimum pairs created for each image is 522:
>   * 87 negative sensation for each positive sensation x 6 positive sensation which also have a parent which is required for our hierarchical loss
>
> > Even with held-out images, the taxonomy, task definitions, and metric are all co-designed, leaving open the possibility of domain overfitting or leakage from description-generation stages.
>
> **Overfitting**
>
> Two results in Table 2 supports generalization of our fine-tuned LLM:
> * We only fine-tune on real ads while we also compute the agreement of our metric with humans on generated ads which are very different as shown in Fig. 6 and 9 in the updated paper. Our metrics achieves high agreement of 0.68 with human annotators on generated images(30% improvement compared to baseline with the highest agreement, VQAScore).
> * We also only fine-tune the model on descriptions generated by InternVL; however, we also evaluate on descriptions generated by QwenVL further supporting generalization to different styles of descriptions. Our metrics with QwenVL description has substantial agreement of 0.80 (45% improvement compared to baseline with the highest agreement, VQAScore) showing the generalization of our metric to different styles of descriptions.
>
> **Description Generation Leakage**
>
> If the performance of EvoSense was the result of sensation information (interpretation) leakage by MLLMs, the 0-shot LLM should have performed as well as our metric. **We compare our metric with the same setup but utilizing the zero-shot LLMs**. Since the exact same descriptions are given to both fine-tuned and zero-shot LLMs, the agreement of zero-shot version should be high as well. However, the **zero-shot version results in negative agreement** highlighting the importance of fine-tuning.
> * We have added qualitative examples of MLLM generated descriptions to appendix - A4 - Figure 13. In the examples, we show that the MLLMs only provide the detailed description for the image without further interpretations of the image itself or the sensation that it evokes. Unfortunately, since we cannot attach images to the comments, we could not include the example in the comment as well.

---

> ### Author Response · Authors · 2025-11-26
> **Response 3/5**
>
> ### [W3] Novelty of Research Direction
>
> > The claim that this is “the first investigation of how ads evoke the senses” is inaccurate. Prior works such as Behavior-LLaVA (ICLR) and related multimodal behavior modeling studies have already explored the link between generated content and human reactions, across a broader behavioral space—including Reddit upvotes, YouTube likes, replay rates, saliency, memorability, and comment patterns. Those efforts provide a more comprehensive foundation for connecting visual content to audience response.
>
> We respectfully disagree; however, we thank the reviewer for mentioning this related-work. While not targeting the same research aspect, it is related to our work, and we have added to our related works.
>
> Prior multimodal behavior-prediction models (e.g., Behavior-LLaVA) study observable behavioral reactions such as likes, upvotes, or memorability.
> These are external outcomes and do not indicate what sensory experiences an image evokes.
> Our work instead focuses on sensations evoked by the image, which can result in different behaviours in different individuals.
> Quoting from the Behaviour-LLAVA paper itself “Humans produce two kinds of behavioral signals upon observing a message:
> perceptual signals and actions as behavior. Perceptual signals, like seeing, touching, and
> hearing, help a receiver primarily sense the world around her, ultimately guiding her actions. Actions
> are how a receiver acts on the outside world.” this part of Behaviour-LLAVA paper indicates the difference between the sensation signals and actions which is the focus of Behaviour-LLAVA.
>
> Moreover, **Behaviour-LLAVA focuses on the side of the receivers' actions** (understanding their action) while **our work is focused on the sender of signals**, the images that evoke the sensations (both generating and understanding).
>
> To the best of our knowledge no prior dataset provides sensory-evocation annotations, nor do existing evaluation methods attempt to predict or quantify sensory cues in images. Behavioral metrics cannot serve as proxies for sensory perception because different users may behave differently despite experiencing similar sensory impressions.
> This establishes sensory evocation as a distinct and previously uninvestigated problem, and our contributions directly address that gap.

---

> ### Author Response · Authors · 2025-11-26
> **Response 4/5**
>
> ### [W4] Persuasion and Sensation Intensity Relation
>
> > A major conceptual weakness is that the paper equates sensation intensity with persuasive effectiveness without providing empirical behavioral validation. While the authors frame sensory evocation as “a persuasive strategy” and “a crucial element of persuasive advertising,” they offer no A/B or user studies (e.g., click-throughs, recall, purchase intent) to substantiate this claim. The link between sensory evocation and actual persuasion therefore remains hypothetical rather than experimentally demonstrated.
>
> * **Marketing Publications**
>
>     We are not discovering the relation between the sensation and persuasion, instead we are building on prior marketing and psychology publications to explore the machine learning techniques in this area. As also referenced in the paper, Sensory Advertisement is a well-studied concept in marketing research. Since this has been previously extensively studied we rely on and trust the outcome of previous marketing and psychology works. The claim is already supported by the following papers, and books from marketing research including the user study.
>
>     Lindstrom in his book [1] cited over 1400 times, shows the relation between the sensory appeal and brand effectiveness, and shows that the brands should use senses based on their products. They further suggest that the sensory perception in marketing results in consumers’ loyalty to the brand.
>
>     [2] cited over 130 times, designs two studies on relation between sensory advertisements and attitude toward the brand. They show sensory ads increase positive attitude toward the brand (increasing the effectiveness of advertisement) and the reason for why sensory advertisement works is that it triggers the self-referencing, i.e.  audience imagining using the product.
>
>     [3] cited over 380 times, discusses how Sensory Marketing influences the effectiveness of the advertisement highlighting improvement in the effectiveness of ads by evoking mental simulation.
>
>     [4] cited over 130 times, reviewed the existing works in sensory images and their effects on consumer behavior, and their findings suggest that sensory advertisements increase the mental imagery in consumers and mental imagery can enhance the ad persuasion.
>
>     [1] [Lindstrom, Martin. "Brand sense: How to build powerful brands through touch, taste, smell, sight and sound." Strategic Direction 22.2 (2006).](https://www.emerald.com/sd/article/doi/10.1108/sd.2006.05622bae.001/353448/Brand-Sense-How-to-Build-Powerful-Brands-Through)
>
>     [2] [Yoon, Sung-Joon, and Ji Eun Park. "Do sensory ad appeals influence brand attitude?." Journal of Business Research65.11 (2012): 1534-1542.](https://www.sciencedirect.com/science/article/pii/S0148296311000701)
>
>     [3] [Krishna, Aradhna, Luca Cian, and Tatiana Sokolova. "The power of sensory marketing in advertising." Current Opinion in Psychology 10 (2016): 142-147.](https://www.sciencedirect.com/science/article/abs/pii/S2352250X16000099)
>
>     [4] [Elder, Ryan S., and Aradhna Krishna. "A review of sensory imagery for consumer psychology." Journal of Consumer Psychology 32.2 (2022): 293-315.](https://myscp.onlinelibrary.wiley.com/doi/full/10.1002/jcpy.1242)
>
> * **Persuasion over Sensation Plot**
>
>     We have also plotted the persuasion score computed using Persuasion metric from [5] over the sensation scores for different generated images. We have added the plot to the Appendix - A1 - Figure 7-d. We observe that the persuasion increases with the increase of sensation intensity.
>
>     [5] [Aghazadeh, Aysan, and Adriana Kovashka. "Cap: Evaluation of persuasive and creative image generation." Proceedings of the IEEE/CVF International Conference on Computer Vision. 2025.](https://openaccess.thecvf.com/content/ICCV2025/html/Aghazadeh_CAP_Evaluation_of_Persuasive_and_Creative_Image_Generation_ICCV_2025_paper.html)

---

> ### Author Response · Authors · 2025-11-26
> **Response 5/5**
>
> ### [W5] Shallow Benchmark
>
> > The T2I benchmarking is shallow. Only five models are tested, with approximately 75 images per model, fixed random seeds, and some quantization. This limited scope is insufficient to draw generalizable conclusions across a rapidly evolving T2I ecosystem, especially given the variation in prompt formats, diffusion steps, and stylistic priors across models.
>
> * We find this in contradiction with second strength point, and would appreciate it if you could explain and resolve the contradiction. In second strength point, the benchmarked is recognized as cohesive with a complete setup and here it is called shallow.
> * We annotated 75 images, the number of images generated by each model in our evaluation setup is 700 for Sensory Ad generation and 960 for Sensory Image Generation.
> * We have annotated 50 more generated images which will release with the previously annotated images upon the acceptance of paper. However, the goal of annotated images is solely for evaluation of our EvoSense metric, and we rely on the metric for evaluation of the models in the benchmark.
> * We agree that with the current rapidly evolving T2I ecosystem, no finding is guaranteed to stay the same. In fact, we believe the big part of the contribution of benchmarks to the field is to find the shortcoming of current methods, to address them and evolve the methods. And we hope with this benchmark we introduce a significantly underexplored research for future research.
>
> ---
>
> ### [W6] Agreement Evaluation
>
> >The paper reports a κ = 0.86 “near-perfect” agreement between EvoSense and human annotators but does not provide confidence intervals, standard errors, or per-sensation breakdowns, which limits interpretability. Comparing κ across model types is statistically problematic because κ depends on class prevalence, which differs between LLM-only and MLLM-based setups. There is also no power or uncertainty analysis to demonstrate that these results are stable given the small dataset al. sensations are aggregated into a single κ/r score, even though some (e.g., “Glow”) are visually trivial while others (e.g., “Dryness”) are abstract and harder to ground. Without stratified or normalized results, it is unclear whether the model’s high κ reflects true generalization or dominance by a few easy classes. Consequently, the claim of “human-level” agreement is numerically strong but statistically under-supported.
>
> Thanks for the suggestion, we have added the per high level sensation break-down for agreement evaluation and confidence intervals for kappa metric (on 100 images and 10000 image-sensation pairs):
>
> |                   Metric                    | touch | smell | sound | taste | sight |   All    | 95% CI |
> |:-------------------------------------------:| :---: | :---: | :---: | :---: | :---: |:--------:| :---: |
> |                  VQAScore                   | 0.58 | 0.60 | 0.42 | 0.65 | 0.58 |   0.57   | [0.561, 0.570] |
> |                  PickScore                  | 0.38 | 0.45 | 0.12 | 0.36 | 0.30 |   0.36   | [0.350, 0.361] |
> |                  CLIPScore                  | 0.48 | 0.47 | 0.36 | 0.41 | 0.30 |   0.44   | [0.430, 0.440] |
> |                Image-Reward                 | 0.49 | 0.50 | 0.38 | 0.34 | 0.45 |   0.46   | [0.457, 0.467] |
> |      EvoSense (LLAMA3-instruct 0-shot)      | -0.09 | 0.08 | -0.22 | -0.005 | -0.007 |  -0.04   | [-0.038, -0.027] |
> |          EvoSense (QwenLM 0-shot)           | -0.15 | 0.04 | -0.22 | 0.03 | 0.003 |  -0.06   | [-0.064, -0.053] |
> |             InternVL as a judge             | 0.54 | 0.48 | 0.43 | 0.54 | 0.49 |   0.50   | [0.507, 0.514] |
> |              QwenVL as a judge              | 0.55 | 0.48 | 0.43 | 0.54 | 0.50 |   0.50   | [0.507, 0.514] |
> | EvoSense (LLAMA3-instruct + $D_{InternVL}$) | **0.79** | **0.82** | **0.77** | **0.84** | **0.85** | **0.80** | [0.806, 0.813] |
> |  EvoSense (LLAMA3-instruct + $D_{QWenVL}$)   | 0.76 | 0.77 | 0.70 | 0.79 | 0.73 |   0.76   | [0.754, 0.761] |
> |     EvoSense (QwenLM + $D_{InternVL}$)      | 0.64 | 0.69 | 0.57 | 0.73 | 0.64 |   0.66   | [0.658, 0.666] |
> |      EvoSense (QwenLM + $D_{QWenVL}$)       | 0.62 | 0.66 | 0.50 | 0.67 | 0.58 |   0.61   | [0.612, 0.621] |
>
> The table, shows that our metric agreement is fairly consistent over different sensations improving at least by 44% compared to baselines.

---

### Official Review · Reviewer_yyYb · 2025-11-11

**Soundness:** 2
**Presentation:** 2
**Contribution:** 2
**Rating:** 4
**Confidence:** 2

**Summary:**

The paper introduces the SensoryAd task to generate an advertisement that evokes specific sensory cues. The authors build a build a sensation taxonomy, curate a dataset of 670 ads sampled from Pitts Ads data and collect crowdsourced annotations about sensory information evoked from these images. Two sensation classification tasks are designed – hierarchical and flat. Further, an evaluation method called EvoSense is presented. It measures how well a text-to-image model evokes a given sensation. Results indicate that the proposed EvoSense model shows better agreement with human annotations compared to existing metrics like ImageReward, CLIP score, Pick Score, etc. Finally, a benchmark of different text-to-image models on the sensory ad generation task is presented.

**Strengths:**

The task of sensory ad generation is well motivated for applications in advertising. This paper attempts to address a gap by studying this important problem in the context of text to image models.
The EvoSense evaluation pipeline is a key contribution. Compared to traditional image evaluation metrics (VQA score, ImageReward, CLIP score, Pick score), EvoSense demonstrates substantially better agreement with human ratings.
The introduction of sensation classification tasks and benchmarking text to image models on sensory ad generation task are well-scoped problems useful to the community.

**Weaknesses:**

Conceptually, the goal of sensory marketing would be to enhance the appeal and persuasiveness of ads to a viewer. The paper does not discuss this connection about how invoking sensations impacts the persuasiveness of ad images. Further, the impact of the paper can be elevated if  the authors can provide evidence through a user study to show the above relation, if there exists any. If such a relation exists, the authors should discuss the ethical implications of sensory ad generation.
The authors report that they trained EvoSense on only 50 images which is quite surprising to me. I wonder if there is enough signal in this data to help the model demonstrate generalisation.
The authors sampled only 670 real ad images from the Pitts Ad dataset which is quite modest in size. It is not clear if the curate subset is representative enough and if there is any selection bias. The Pitts Ad data covers various ad types (eg: vintage vs modern) , industry sectors, etc. Can you please report the properties of the sampled version along these axes?
While the authors report the agreement of EvoSense with human ratings but details about inter-annotator agreement in the data seem missing. That would serve as a topline reference for interpreting the results of any evaluation method.
While EvoSense achieves high agreement with human ratings (κ = 0.86), however the Pearson correlation for this task is still a bit low (r = 0.38). Do these observations suggest that EvoSense might be good at picking the most relevant sensation but less so for actually predicting the intensity score of the sensations? I would like to see some more analysis being provided to explain this observation.

**Questions:**

Please see the weakness section and provide additional details as requested.

---

> ### Author Response · Authors · 2025-11-26
> **Response 1/3**
>
> We thank the reviewer for the constructive feedbacks and the opportunity to respond and resolve the concerns.
> We are thrilled that you found our tasks well motivated, well scoped, and useful to the community. We are encouraged that you found our EvoSense metric a key contribution.
> In the next comments we address the weaknesses mentioned in the review.
>
> ---
> ### [W1] Connection between sensation evocation and persuasion
> > Conceptually, the goal of sensory marketing would be to enhance the appeal and persuasiveness of ads to a viewer. The paper does not discuss this connection about how invoking sensations impacts the persuasiveness of ad images. Further, the impact of the paper can be elevated if the authors can provide evidence through a user study to show the above relation, if there exists any.
> * **Marketing Publications**
>
>     We are not discovering the relation between the sensation and persuasion, instead we are building on prior marketing and psychology publications to explore the machine learning techniques in this area. As also referenced in the paper, Sensory Advertisement is a well-studied concept in marketing research. Since this has been previously extensively studied we rely on and trust the outcome of previous marketing and psychology works.
>     The claim is already supported by the following papers, and books from marketing research including the user study.
>
>     Lindstrom in his book [1] cited over 1400 times, shows the relation between the sensory appeal and brand effectiveness, and shows that the brands should use senses based on their products. They further suggest that the sensory perception in marketing results in consumers’ loyalty to the brand.
>
>     [2] cited over 130 times, designs two studies on relation between sensory advertisements and attitude toward the brand. They show sensory ads increase positive attitude toward the brand (increasing the effectiveness of advertisement) and the reason for why sensory advertisement works is that it triggers the self-referencing, i.e.  audience imagining using the product.
>
>     [3] cited over 380 times, discusses how Sensory Marketing influences the effectiveness of the advertisement highlighting improvement in the effectiveness of ads by evoking mental simulation.
>
>     [4] cited over 130 times, reviewed the existing works in sensory images and their effects on consumer behavior, and their findings suggest that sensory advertisements increase the mental imagery in consumers and mental imagery can enhance the ad persuasion.
>
>     [1] [Lindstrom, Martin. "Brand sense: How to build powerful brands through touch, taste, smell, sight and sound." Strategic Direction 22.2 (2006).](https://www.emerald.com/sd/article/doi/10.1108/sd.2006.05622bae.001/353448/Brand-Sense-How-to-Build-Powerful-Brands-Through)
>
>     [2] [Yoon, Sung-Joon, and Ji Eun Park. "Do sensory ad appeals influence brand attitude?." Journal of Business Research65.11 (2012): 1534-1542.](https://www.sciencedirect.com/science/article/pii/S0148296311000701)
>
>     [3] [Krishna, Aradhna, Luca Cian, and Tatiana Sokolova. "The power of sensory marketing in advertising." Current Opinion in Psychology 10 (2016): 142-147.](https://www.sciencedirect.com/science/article/abs/pii/S2352250X16000099)
>
>     [4] [Elder, Ryan S., and Aradhna Krishna. "A review of sensory imagery for consumer psychology." Journal of Consumer Psychology 32.2 (2022): 293-315.](https://myscp.onlinelibrary.wiley.com/doi/full/10.1002/jcpy.1242)
>
>
> * **Persuasion over Sensation Plot**
>
>     We have also plotted the persuasion score computed using Persuasion metric from [5] over the sensation scores for different generated images. We have added the plot to the Appendix - A1 - Figure 7-d. We observe that the persuasion increases with the increase of sensation intensity.
>
>     [5] [Aghazadeh, Aysan, and Adriana Kovashka. "Cap: Evaluation of persuasive and creative image generation." Proceedings of the IEEE/CVF International Conference on Computer Vision. 2025.](https://openaccess.thecvf.com/content/ICCV2025/html/Aghazadeh_CAP_Evaluation_of_Persuasive_and_Creative_Image_Generation_ICCV_2025_paper.html)

---

> ### Author Response · Authors · 2025-11-26
> **Response 2/3**
>
> ### [W2] Ethical Implications
> > If such a relation exists, the authors should discuss the ethical implications of sensory ad generation.
>
>  There are two main implications:
>    * First, in overall the advertisement (especially commercial advertisements) are not always following ethical goals. However, **this is a general problem with any T2I model generating persuasive content.**
>    * Second, the model might generate sensitive content (e.g. pain sensation in advertisement) for a certain group of audience and this is one of the motivations for classification tasks. While automatically generating the Sensory Ads can be helpful, some sensitive sensations should be detected and prevented from showing to a specific group of audience. This is why it is also important to be able to classify the sensations evoked by the image.
>
>  We have added these implications as the ethical implication discussion in the Appendix - A2 section.
>
> ---
>
> ### [W3] Number of Images in the Fine-tuning Set
> > The authors report that they trained EvoSense on only 50 images which is quite surprising to me.
>
> While there are only 50 images (now we also fine-tune on 100 images in new experiments), each image is paired with negative and positive sensation, which expands the data to 21000 samples that the models have been fine-tuned on.
>
> **Number of image-(positive, negative, parent of positive) pairs**
>   * Each image is annotated with a maximum of 9 sensations that the image evokes.
>   * Each sensation is associated with a score on the quality of the sensation evocation.
>   * For each image we collect the (positive - negative) sensation pair for each two sensations:
>     1. If the score of one is higher than the other, then the sensation with higher score is positive and the one with lower score is negative sensation.
>     2. If the scores are equal then they are not paired together.
>
>   * This way each sensation might be positive in one pair and negative in another.
>   * Given that there are 96 sensations in our taxonomy, and maximum of 9 sensations are only selected to be evoked the minimum pairs created for each image is 522:
>     * 87 negative sensation for each positive sensation x 6 positive sensation which also have a parent which is required for our hierarchical loss
>
> Also, results on classification task in Table 1 represents that the precision of LLMs/MLLMs is much lower than recall, suggesting the rejecting the incorrect sensations is the main issue of these models, and our fine-tuning setup teaches the model to reject the negative sensations by first pairing each positive sensation with 95 negative sensations and utilizing Contrastive Preference Optimization.
>
> **Generalization of EvoSense:**
>
> > I wonder if there is enough signal in this data to help the model demonstrate generalisation.
>
> **Results in Table 2 Supporting EvoSense Generalization**
>
> Two results supports generalization of our fine-tuned LLM:
> * We only fine-tune on real ads while we also compute the agreement of our metric with humans on generated ads which are very different as shown in Fig. 6 and 9 in the updated paper. Our metrics achieves high agreement of 0.68 with human annotators on generated images (30% improvement compared to baseline with the highest agreement, VQAScore).
> * We also only fine-tune the model on descriptions generated by InternVL; however, we also evaluate on descriptions generated by QwenVL further supporting generalization to different styles of descriptions. Our metrics with QwenVL description has substantial agreement of 0.80 (45% improvement compared to baseline with the highest agreement, VQAScore) showing the generalization of our metric to different styles of descriptions.
>
> **Fine-tuning LLMs on more data**
>
> We also, increased number of steps to 40000 and the number of images to 100 and here are the new results showing consistency in the agreement of the model with human annotations over different number of fine-tuning steps:
>
>  |                 Metric                 | #steps | touch | smell | sound | taste | sight | All |
>  |:--------------------------------------:| :--- | :--- | :--- | :--- | :--- | :--- | :--- |
>  | EvoSense (LLAMA3-instruct + $D_{InternVL}$) | **21000** | 0.79 | 0.82 | 0.77 | 0.84 | 0.85 | 0.80 |
>  | EvoSense (LLAMA3-instruct + $D_{InternVL}$) | **25000** | 0.80 | 0.82 | 0.78 | 0.83 | 0.88 | 0.81 |
>  | EvoSense (LLAMA3-instruct + $D_{InternVL}$) | **30000** | 0.80 | 0.82 | 0.78 | 0.84 | 0.88 | 0.81 |
>  | EvoSense (LLAMA3-instruct + $D_{InternVL}$) | **40000** | 0.80 | 0.81 | 0.78 | 0.84 | 0.87 | 0.80 |
>
> While the results are consistent over different fine-tuning steps in the table, our initial experiments with few number of images (upto 10 images) showed low agreement with human annotations suggesting there is a minimum number of images/steps required to reach high agreement with human.

---

> ### Author Response · Authors · 2025-11-26
> **Response 3/3**
>
> ### [W4] Dataset Size
> >The authors sampled only 670 real ad images from the Pitts Ad dataset which is quite modest in size.
>
> Due to the cost issue and annotation costs, we introduce the evaluation dataset for sensory advertisement. From PittAd dataset we randomly select 670 (**from both Commercial and Public Service Advertisements**).
>
> > It is not clear if the curate subset is representative enough and if there is any selection bias.
>
> We selected 670 images randomly covering 95 sensation categories and over 40 topics.
> We have added the topic and sensation diversity in appendix - A.3 - Fig. 11. The figure represent the distribution of images over 10 most frequent topics and 5 sensations.
>
> The following tables show the diversity of images over 10 most frequent topics, and 5 high level sensations.
>
> **Topic distribution:**
>
> |               Topic               |  # Images  |
> |:---------------------------------:|:----------:|
> |   Beauty products and cosmetics   |    251     |
> |     Clothing and accessories      |    136     |
> |             Shopping              |     75     |
> |           Human rights            |     70     |
> |            Self esteem            |     61     |
> |              Smoking              |     52     |
> |         Domestic violence         |     42     |
> |           Animal rights           |     41     |
> |          Media and arts           |     37     |
> |              dating               |     34     |
> |  Sports equipment and activities  |     32     |
> |    Healthcare and medications     |     31     |
> |              Alcohol              |     26     |
> |       Political candidates        |     26     |
> |            Electronics            |     23     |
>
> **Sensation diversity:**
>
> | Sensation  | %images evoking the sensation  |
> |:----------:|:------------------------------:|
> |   touch    |             40.14              |
> |   sound    |             26.44              |
> |   smell    |             31.73              |
> |   taste    |             12.98              |
> |   sight    |             19.71              |
> |    none    |             28.94              |
>
> ---
>
> ### [W5] Human-Human Agreement
> > While the authors report the agreement of EvoSense with human ratings but details about inter-annotator agreement in the data seem missing. That would serve as a topline reference for interpreting the results of any evaluation method.
>
> We compute the human - human agreement on 60 images (5760 image-sensation pairs) and the kappa agreement between the human annotators is **0.83 with 95% CI = [0.831, 0.838]**. We have added the human-human agreement to the updated paper.
>
> ---
>
> ### [W6] Kappa vs Pearson
> > While EvoSense achieves high agreement with human ratings (κ = 0.86), however the Pearson correlation for this task is still a bit low (r = 0.38)
>
> * The difference **(as also reflected on baseline metrics)** is because the annotators choose up-to 3 sensation groups evoked by the image, and the rest of the scores are 0. On the other hand, the computational metrics (including EvoSense and the baselines) choose different scores for each sensation. For computing κ agreement, we use the sensation intensity as the criteria for choosing the winner sensation for the image given each two sensation. We ignore the sensation pairs where the human annotators assign the same score to both sensations.
> This way we significantly reduce the sparsity of human annotations for the image. So, while the incorrect sensations are included paired with selected sensations, they are not included as paired with other unselected sensations. This is why κ is bigger than r.
> * We have added an example sensation score for an image to the Appendix - A5 - Figure 12, showing the problem of correlation because of the **sparsity of the human scores**. The figure represents while high scores assigned by metric represent the sensations evoked by the image selected by the human, because of the sudden drop in the values of human scores, correlation becomes lower.

---

### Author Response · Authors · 2025-11-26
**Summary**

We thank the reviewers for their constructive and insightful comments. We are happy that reviewers found our research direction “genuinely novel” and our analysis both systematic and insightful, especially in how we evaluate and understand sensory advertisements (Reviewers FpSs, S1yq). We appreciate the recognition of our tasks as well motivated, well scoped, well-defined, and useful to community forming “usable foundations for future research” (Reviewers yyYb, FpSs). We also thank them for highlighting the cohesiveness and usefulness of our benchmarks serving “as a cornerstone for future research” (Reviewers FpSs, S1yq). We are pleased that the EvoSense metric was regarded as a key contribution that can advance evaluation in this space (Reviewer yyYb). Finally, we appreciate the positive remarks on the idea of exploring sensory visual content (Reviewer dfxp).

We thank the reviewers again for their thoughtful comments and will address their constructive suggestions in detail as a comment to each review. Here are the summary of new experiments and analysis:
* Annotation and Data Reliability
  * Human - human agreement on data annotation
  * Annotators and annotation process details
  * Data diversity
* EvoSense Evaluation
  * Human - metric agreement on more images
  * Reporting confidence-interval for agreement scores
  * Adding MLLMs as a judge for sensation intensity to the baselines
  * Per sensation break-down for human - metric agreements
  * Training-step ablation: increasing the number of train-steps in the fine-tuning of LLM to include more images
* Sensory Ad Generation Benchmarking
  * Quantitative analysis of exaggeration in sensation evocation
  * Alignment of images and advertisement message added to analysis
  * T2I Model Fine-tuning Ablation: Fine-tuned SD3 on Sensory Ad generation task

---

### Meta-Review · Area_Chair_WqzS · 2026-01-05

**Summary:**

This paper is reviewed by four reviewers. All reviewers think this paper currently has some flaws and make a reject reocmmendation. Although it seems that some of comments are generated by LLM, some of them are reasonable. The authors provide detailed responses to these comments. Some of confusion points are clarified on the technical details and dataset details. Yet, some critical points remain, such as small size of proposed dataset, missing study between sensation evocation and persuasion, small number of images used to fine tune the model, missing technical novelty on the method etc. These critical points require more study and experiments to fully address. Thus, the meta reviewer thinks this paper is not ready for publication and makes a reject recommendation.

**Reviewer Concerns:**

Some of reviewers concern on details are clarified. Yet, some critical points still remain such as small size of proposed dataset, missing study between sensation evocation and persuasion, small number of images used to fine tune the model, missing technical novelty on the method. The meta reviewer suggest the authors to perform experiments on much larger scale.

**Reviewer Scores:**

For four reviewers, I think all of them could change their review score a little bit. But, it is really hard for them to increase the score higher than 5.

---

### Decision · Program_Chairs · 2026-01-26

Reject